# Combined SMAP/SMOS Thin Sea Ice Thickness Retrieval

Cătălin Pațilea[1], Georg Heygster[1], Marcus Huntemann[2,1], and Gunnar Spreen[1]

[1]Institute of Environmental Physics, University of Bremen, Bremen, Germany
[2]Alfred Wegener Institute, Bremerhaven, Germany

*Correspondence to:* Cătălin Pațilea (cpatilea@iup.physik.uni-bremen.de)

**Abstract.** The spaceborne passive microwave sensors Soil Moisture Ocean Salinity (SMOS) and Soil Moisture Active Passive (SMAP) provide brightness temperature data at L-band (1.4 GHz). At this low frequency the atmosphere is close to transparent and in polar regions the thickness of thin sea ice can be derived. SMOS measurements covers a large incidence angle range whereas SMAP observes at a fixed 40° incidence angle. By using brightness temperatures at a fixed incidence angle obtained directly (SMAP), or through interpolation (SMOS), thin sea ice thickness retrieval is more consistent as the incidence angle effects do not have to be taken into account. Here we transfer a retrieval algorithm for thickness of thin sea ice (up to 50 cm) from SMOS data at 40° to 50° incidence angle to the fixed incidence angle of SMAP. The SMOS brightness temperatures (TBs) at a given incidence angle are estimated using empirical fit functions. SMAP TBs are calibrated to SMOS for providing a merged SMOS/SMAP sea ice thickness product. The new merged SMOS/SMAP thin ice thickness product improved in several ways compared to previous thin ice thickness retrievals: (i) The combined product provides a better temporal and spatial coverage of the polar regions due to the usage of two sensors. (ii) The RFI filtering method was improved, which results in higher data availability over both ocean and sea ice areas. (iii) For the inter-calibration between SMOS and SMAP brightness temperatures the root mean square difference (RMSD) got reduced by 30% relative to a prior attempt. (iv) The algorithm presented here allows also for separate retrieval from any of the two sensors, which makes the ice thickness dataset more resistant against failure of one of the sensors. A new way to estimate the uncertainty of ice thickness retrieval was implemented, which is based on the brightness temperature sensitivities.

## 1   Introduction

Sea ice is an important climate parameter (Moritz et al., 2002; Stroeve et al., 2007; Holland et al., 2010) and accurate knowledge of sea ice properties is needed for weather and climate modeling and prediction and for ship routing. The thickness of the ice is one of the parameters that determines the resistance against the deforming forces of wind and ocean currents (Häkkinen, 1987; Yu et al., 2001). Even a thin layer of sea ice inhibits evaporation, reduces heat and gas exchange between ocean and atmosphere and increases the albedo (Maykut, 1978; Perovich et al., 2012). Sea ice also provides a solid surface for snow to deposit, which further reduces heat exchange and increases albedo (Shokr and Sinha, 2015).

The Soil Moisture Ocean Salinity (SMOS) satellite was launched by ESA in November 2009. It is a synthetic aperture passive microwave radiometer working at L-band (1.4 GHz). The aperture synthesis requires an array of small antennas reducing the

total weight and size of the satellite. The instrument works in a full polarimetric mode, recording all four Stokes parameters. Its large field of view allows for multi-angular observations organized in approximately 1200 km × 1200 km snapshots.

SMOS has been developed for retrieving soil moisture (Kerr et al., 2012), by inferring the surface emissivity which is correlated with the moisture content, and sea surface salinity (Zine et al., 2008; Font et al., 2010), where the measured brightness temperatures (TB) are linked with the sea salinity through the dielectric constant of the water in the first few centimeters. Modeling and observations showed that at this frequency the radiation is sensitive to ice thickness up to 50 cm (Kaleschke et al., 2010, 2012). The atmosphere has little influence on the radiation at L-band as both absorption and scattering are small (Skou and Hoffman-Bang, 2005). The correlation of ice thickness with emitted radiation together with a small atmospheric contribution make SMOS a candidate for thickness retrieval of thin sea ice. To date, two sea ice thickness retrieval algorithms have been developed for SMOS, one using the TB intensity averaged over incidence angles between 0° and 40° (Tian-Kunze et al., 2014) and one using intensity and polarization difference averaged over incidence angles between 40° and 50° (Huntemann et al., 2014).

In 2015 the Soil Moisture Active Passive (SMAP) satellite was launched by NASA (Entekhabi et al., 2010, 2014). It carries two sensors on board, an L-band radiometer, and a radar which share a rotating 6 m real aperture antenna reflector. The radar was recording high resolution (1 to 3 km) data used for soil moisture sensing, until it failed after three months. In contrast to the synthetic aperture observations of SMOS, the real aperture antenna observations of SMAP cover an area of 36 km × 47 km at a fixed incidence angle of 40° and results in a swath with an approximate width of 1000 km. The preceding technical details of SMAP were presented in Entekhabi et al. (2014). SMAP also includes on board detection and filtering of radio frequency interference (RFI) while SMOS does not (Mohammed et al., 2016).

After the launch of SMAP, different approaches were taken to convert data products between the two sensors. A previous approach to convert SMOS to SMAP TBs for usage in soil moisture retrieval and assimilation systems is presented in Lannoy et al. (2015) and involves a quadratic fitting of the SMOS TBs at the SMAP incidence angle and employing auxiliary data and an empirical atmospheric model to correct for the atmospheric and extraterrestrial contributions, respectively. In contrast, Huntemann et al. (2016) converts SMAP 40° surface TBs to SMOS top of the atmosphere equivalent 40 to 50° averaged TBs through two linear regressions. A more recent attempt for inter-calibrating SMOS and SMAP data, and using the resulting TBs for a separate SMAP, but also a combined SIT retrieval was presented in Schmitt and Kaleschke (2018).

In this article, we present a combined Sea Ice Thickness (SIT) dataset using input from both sensors by calibrating the SMAP TBs to those of SMOS (Sect. 4). As a first step, an inter-calibration of the TBs of the two sensors is required due to a possible warm bias in SMOS data (Sect. 2) and due to corrections for galactic noise and sun specular reflection contained in the SMAP but not in the SMOS TB data. In addition, the SIT retrieval from Huntemann et al. (2014) is adapted to the new version 6.20 of the SMOS Level 1C data and it will be used as a reference for all other comparisons (Sect. 3.1). This new retrieval is combined with a fit function for the dependence of horizontal and vertical TBs (from now on referred as $TB_h$ and $TB_v$, respectively) on the incidence angle (Sect. 3.2). The fit function is used for RFI filtering and for SIT retrieval at a fixed incidence angle. The fit is also a step required for the SMOS and SMAP merged product to combine the observations of the two sensors at a common incidence angle.

## 2 SMOS and SMAP data sources

The MIRAS radiometer onboard the SMOS satellite has 69 receivers on three arms measuring radiances at 1.4 GHz (Kerr et al., 2001). One complete set of data from the aperture synthesis process done each 1.2 seconds is called a snapshot. For this investigation the SMOS Level 1C (L1C) ocean data gridded on the icosahedron Snyder equal area (ISEA) 4H9 grid (Sahr et al., 2003) is used. The grid spacing is 15 km while the SMOS footprint size varies with incidence angle from approximately 30 km×30 km at nadir to 90 km×30 km at 65° (Castro, 2008). Over the whole field of view the average resolution is approximately 43 km. The Level 1C data is provided within 24 h of acquisition.

In full polarization mode, all four Stokes parameters are measured. Data is recorded in the reference plane of the antenna as $T_X$, $T_Y$, $T_3$ and $T_4$, and is converted to $TB_h$, $TB_v$, $TB_3$ and $TB_4$ in the Earth surface plane (Zine et al., 2008) using

$$
\begin{bmatrix} T_X \\ T_Y \\ T_3 \\ T_4 \end{bmatrix} = \begin{bmatrix} \cos^2(\alpha) & \sin^2(\alpha) & -\cos(\alpha)\sin(\alpha) & 0 \\ \sin^2(\alpha) & \cos^2(\alpha) & \cos(\alpha)\sin(\alpha) & 0 \\ \sin(2\alpha) & -\sin(2\alpha) & \cos(2\alpha) & 0 \\ 0 & 0 & 0 & 1 \end{bmatrix} \begin{bmatrix} TB_h \\ TB_v \\ TB_3 \\ TB_4 \end{bmatrix}, \tag{1}
$$

where $\alpha = \alpha_{gr} + \omega_{F_r}$, $\alpha_{gr}$ is the georotation angle and $\omega_{F_r}$ is the Faraday rotation angle. Within a snapshot just one or two of the Stokes parameters are measured at the same time. When only one of the Stokes parameters is measured, all three arms of the sensor record the same polarization. In the case of recording a cross-polarized snapshot, one arm of the sensor records one polarization while the other two record the other polarization (McMullan et al., 2008). Measurements of single ($XX$ or $YY$) and cross-polarization (($XX$,$XY$) or ($YY$,$XY$)) are done alternatively. In order to obtain the values for $TB_h$ and $TB_v$ from the matrix, depending if the current measurement is single or cross-polarization, we have to use one or two adjacent snapshots. The missing values required for the conversion are interpolated from neighboring snapshots within a 2.5 s range and with a maximum incidence angle difference between the measurements of 0.5°.

The SMOS L1C version 6.20 has been operationally available since 5 May 2015 and also older acquisitions were reprocessed. This version adds better RFI flagging and improves the long-term and seasonal stability of the measurements. At the same time it introduces a warm bias in the TBs of approximately 1.4 K relative to the previous version 5.05 over ocean. The bias over the ocean can be 1 K too warm with respect to the true values. Over Antarctica and land, the bias is above 2 K, which is closer to modeled and ground based measurements. The new version also reduces the difference in TB between ascending and descending overflights over ocean at low latitudes. At high latitudes such changes were not documented. Before, the difference varied considerably with time and latitude due to thermal variations in the instrument. All of the technical details described above about the new data version are presented in SMOS Calibration team and Expert Support Laboratory Level 1 (2015)

The SMAP satellite is positioned on a quasi-polar sun-synchronous orbit with an ascending equator crossing time at 6 pm, while SMOS has an equator crossing time at 6 am. SMAP carries a conically scanning radiometer with a fixed incidence angle of 40° which leads to a narrower swath and decreases the area covered at the pole compared to SMOS. The footprint of a SMAP observation is approximately 36 km × 47 km, resulting in an approximate resolution of 40 km. In this study, the

SMAP Level 1B data is used which contains time ordered ungridded Top Of the Atmosphere (TOA) TBs. It is available from 31 March 2015 and is provided with a latency of about 12 h.

SMOS and SMAP observe in a restricted band (1.400-1.427 GHz) reserved for passive radioastronomical use. Nevertheless, there are surface based artificial sources causing RFI (Mecklenburg et al., 2012). The image reconstruction process required to obtain the SMOS TBs includes an inverse Fourier transform (Corbella et al., 2004). Therefore, not only the grid cells that contain the RFI source are affected but the whole snapshot can be contaminated, resulting in high or even negative TBs (Oliva et al., 2012). Since in nature TB will not exceed 300 K over the polar ocean (Kaleschke et al., 2010; Mills and Heygster, 2011; Tian-Kunze et al., 2014), a simple RFI filter is used to eliminate the whole snapshot which contains at least one TB exceeding this threshold. This filter is used in the SIT retrieval algorithm presented in Huntemann et al. (2014). An alternative approach for filtering RFI has been shown in Huntemann and Heygster (2015) where incidence angle binning is used, resulting in a higher preservation of data and fewer gaps on the grid. In this paper we use a new iterative method based on the removal of data with high difference relative to the a SMOS TBs fit curve, as presented in Sect. 3.2. Since SMAP contains onboard hardware for detection and filtering of RFI and neighboring pixels are unaffected by an RFI source, no additional filtering is required for the SMAP Level 1B data.

## 3 Sea ice thickness retrieval using a fit function

Due to the new SMOS data version 6.20 used here compared to version 5.05 used in Huntemann et al. (2014), a retraining of the SMOS thin ice thickness retrieval is necessary. First, in Sect. 3.1 we use the method presented in Huntemann et al. (2014) just using the newer data version 6.20. This involves averaging the TBs between 40 and 50° incidence angle. Secondly, we employ a fitting function using the dependence of TB on incidence angle (Section 3.2) as input for the retrieval (Section 3.3). The fitting function is used to obtain SMOS TBs at a fixed incidence angle.

### 3.1 SMOS retrieval retraining

Three SMOS grid cells in the Kara and Barents Sea located at (78.71°N,57.41°E),(77.37°N,81.71°E) and (75.81°N,79.57°E) were used for training over a period of three months (1 October - 26 December 2010) with SIT obtained using the relation with the Cumulated Freezing Degree Days (CFDD) based on NCEP temperature data as presented in Huntemann et al. (2014). CFDD is the daily average temperature below -1.8° (freezing point of sea water) integrated over the time with sub freezing temperatures (Bilello, 1961). The relation beween the CFDD and the thickness as presented in Bilello (1961) is $SIT[cm] = 1.33 \cdot (CFDD[°C])^{0.58}$. The ASI (Spreen et al., 2008) Sea Ice Concentration (SIC) product was used to filter low SIC data during the training period. Only during the early part of the freeze-up when ice is really thin the SIC was allowed to have a value between 0-100% (Huntemann et al., 2014) otherwise 100% SIC was required. The TBs are averaged daily over the incidence angle range between 40° to 50°. The functions

$$I_{abc}(x) = a - (a - b) \cdot \exp(-x/c),$$
$$Q_{abcd}(x) = (a - b) \cdot \exp(-(x/c)^d) + b,$$

(2)

are fitted to the intensity $I$ and polarization difference $Q$ data measured over the training areas and the SIT resulting from the CFDD method, where $a, b, c$ and $d$ represent the curves parameters (Table 1), $x$ is the SIT while $I$ and $Q$ are the TB intensity and polarization difference, respectively. The SIT retrieval curve is the result of using the two fitted functions from Equation 2 in the $(Q, I)$ space. For each pair of $Q$ and $I$ the minimum Euclidean distance to the retrieval curve is used to determine the SIT. The retrieval curve parameters for data version 5.05 presented in Table 1 are updated values of the Huntemann et al. (2014) that are currently used for daily processing at the University of Bremen (www.seaice.uni-bremen.de).

Figure 1 shows the retrieval curves in the $(Q, I)$ space. The dots on the curves represent the SIT increasing with intensity and decreasing with polarization difference in steps of 10 cm from 0 cm to 50 cm. Over 50 cm the retrieval is too sensitive to small changes in intensity and polarization difference and it will be cut off. The SIT retrieval curve for data version 5.05 and the retrained curve using the 6.20 data version are shown in black and blue, respectively. The new data version exhibits a value ~1.7 K higher value at zero SIT for intensity and polarization difference. The discrepancy increases up to 3 K at 50 cm SIT.

Figure 2 shows the intensity (left) for 29 October 2010 using daily mean TBs for each grid cell. The data has been regridded to the NSIDC polar stereographic grid with a resolution of 12.5 km. This resolution is an oversampling of the true resolution of SMOS which is 43 km on average. The original validated retrieval (Huntemann et al., 2014) was trained with the old data version and is used as a reference here. The warm bias of the new version is seen in the difference plot (Fig. 2 right) both over ocean area and sea ice. In regions of high contrast like the ice edge or coastlines, both versions tend to produce spillover effects (SMOS Calibration team and Expert Support Laboratory Level 1, 2015). The spillover produces an erroneous increase in TB over ocean areas adjacent to coastlines or ice edge, or decrease in TB over the sea ice near the ice edge. The erroneous values vary between 1 K to 1.5 K (SMOS Calibration team and Expert Support Laboratory Level 1, 2015) in the areas mentioned (not visible in the plot). The errors in TB appear due to calibration errors in the SMOS instrument and systematic spacial ripples (Corbella et al., 2015; Martín-Neira et al., 2016; Li et al., 2017) originating from the Fourier reconstruction of the snapshot (Corbella et al., 2005).

The algorithm trained with SMOS data version 5.05 has been compared with the one trained with version 6.20 for the period 1 October to 26 December 2010, considering SIT from 1 cm to 50 cm. The mean difference of the new retrieval is -0.22 cm while the RMSD is 1.35 cm. From a total of 5.1 million cumulated data points over the 87 days period and 50 cm SIT range, 97% have at most a 3 cm difference. The mean difference and RMSD are below $\pm 1$ cm and 2 cm, respectively, for ice thicknesses below 25 cm. For 50 cm thickness the mean difference increases to $+4$ cm while the RMSD reaches 11 cm.

A test is done to estimate the error introduced by the use of of the original retrieval (Huntemann et al., 2014) with the 6.20 data version. The two algorithms trained with the different data versions take as input the 6.20 data only. The dataset covers the freeze-up period from 1 October to 26 December 2010. The mean difference between the retrained retrieval and the original one is 0.33 cm with 99% of the data having a difference of 3 cm or less, while the RMSD is 0.91 cm. This means that although it is recommended to use the algorithm adapted for the new data version, the difference is below 1 cm thickness on average for SIT below 51 cm if processed with the old algorithm.

## 3.2 SMOS TBs fit characteristics

In the previous section, we have shown that the SIT output with the new data version and new retrieval is consistent with the old data version and retrieval. In all of the next sections the SMOS Level 1C 6.20 version will be used, and when making reference to the original daily mean SIT retrieval, the retrained 6.20 version algorithm from Sect. 3.1 will be used. In each grid cell, the number of data points and the covered incidence angle range are highly variable due to the orbit characteristics, the large incidence angle range of $0°$ to $65°$, and the complex distribution of incidence angle within a SMOS snapshot. Grid cells located closer to the center of the swath will cover a large incidence angle range. Near the swath edges, the range is reduced and low incidence angles are not covered (Font et al., 2010). The snapshots removed using the over 300 K RFI filter can create a local bias in the average incidence angle. The existence of an RFI source before an observed grid cell, relative to the trackline, will result in the elimination of snapshots with high incidence angle data points for that cell. Conversely, an RFI source located after the grid cell of interest will result in elimination of the low incidence angle data points. The varying angle distribution depending on the position in the swath and the data removal due to the RFI filtering for one grid cell may shift the average incidence angle of the ensemble of observations between $40°$ and $50°$ away from the assumed average of $45°$. The average TBs and SIT values retrieved from the affected grid cells will be shifted accordingly. This error can be avoided by fitting a curve to the angular dependent TBs, allowing for a retrieval which uses TBs estimated for a fixed incidence angle.

Here we propose as a solution a modified version of the fit functions described in Zhao et al. (2015). The fit is applied separately to each polarization, horizontal and vertical, for each grid cell using daily observations. An initial filtering of RFI is done by removing observations which are flagged in Level 1C data for either being affected by tails of point source RFI or for indicating RFI by the system temperature standard deviation exceeding the expected trend (Indra Sistemas S.A., 2014). The flagged data is removed before the TBs are transformed from the antenna to the earth reference frame.

The fit functions that describe the dependence of $TB_h$ and $TB_v$ on the incidence angle are

$$
\begin{aligned}
TB_h(\theta) &= a_h \cdot \theta^2 + \frac{C}{2} \cdot [b_h \cdot \sin^2(\theta) + \cos^2(\theta)] \\
TB_v(\theta) &= a_v \cdot \theta^2 + \frac{C}{2} \cdot [b_v \cdot \sin^2(d_v \cdot \theta) + \cos^2(d_v \cdot \theta)].
\end{aligned}
\tag{3}
$$

where $\theta$ represents the incidence angle, $C/2$ is the intensity at nadir, $a_h, b_h, a_v, b_v$ and $d_v$ are five additional parameters used to fit the curves. The Brewster angle effect on the vertically polarized TBs is represented by the additional parameter $d_v$. The fit is done iteratively with a maximum of five steps. For each step the parameter $C$ (Eq. 3) is determined for a given grid cell by first summing up the TBs of horizontal and vertical polarization for each individual observation and then taking the median of the result. The median is used so that any RFI influenced outliers will not influence $C$. Due to asymmetric change in TB between horizontal and vertical polarization at higher incidence angles, only grid cells with at least one observation under $40°$ are considered. This increases the stability of the fit since $C/2$ represents the intensity at nadir. The $40°$ threshold is selected due to increased asymmetry between vertical and horizontal TBs at higher incidence angles which will generate a bias in the computation of the parameter $C$. The other five fit parameters $a_h, b_h, a_v, b_v$ and $d_v$ in the fit functions are determined by a least squares procedure.

At each iteration of the fitting procedure if the RMSD of the fit is higher than 5 K or if the RMSD fit difference between successive iterations exceeds 1 K, 20% of the observations with the highest absolute difference from the fit are removed. After the removal of data, in the next iteration the computation of C and the least squares method to fit the parameters is repeated. The data removal in the iterative process is the second step used to discard possible RFI influences.

At the last iteration, if the RMSD of the fit is higher than 5 K or the RMSD fit difference relative to the fourth iteration is higher than 1 K, the fit parameters will still be used for computation of TBs, at the desired incidence angle, but with a higher RMSD. In the case of non convergence of the least squares procedure for the fit parameters, the grid cell will be discarded from TB computation.

The fit function is not optimized for extrapolation of the covered incidence angle range. Incidence angles not covered by the observations will have high uncertainty. To avoid extrapolation, only grid cells which contain observations with incidence angles both below and above the desired incidence angle are used for the retrieval, e.g. for a reference angle of 45° observations below and above 45° need to be present in the respective grid cell.

A similar approach for fitting SMOS L1C TBs to a fixed incidence angle using the method presented in Zhao et al. (2015) was done in Schmitt and Kaleschke (2018). For filtering RFI it uses RFI flags within the SMOS data, similar to what is done in this study. As a second step, however, they remove whole snapshots if one data point within the snapshot contains a TB value over 300 K. This was also done previously in Huntemann et al. (2014). For this study, however, we introduced an iterative method to fit the brightness temperatures, which does not need a fixed cut-off value for brightness temperature removal anymore. As a result, more data will be removed before the fitting procedure in Schmitt and Kaleschke (2018) compared to the method presented here.

## 3.3 Sea ice thickness retrieval training using fitted data

The retrieval algorithm is retrained as described in Sect. 3.1 but instead of using TBs averaged over 40-50° incidence angle, we use TBs from the fit process (Sect. 3.2) at a nominal incidence angle of 45°. The resulting retrieval curve (Fig. 1 green) has 1.3 K higher polarization difference at 0 cm ice thickness than the algorithm trained with the daily mean data (Fig. 1 blue). The difference decreases to 0.1 K at 20 cm thickness and increases to approximately 0.5 K at 50 cm. This can come from variability in the mean incidence angle. The daily averaged observations have an incidence angle bias of -0.5° (with single differences as high as -2.5°) relative to the assumed 45° one. The smaller incidence angle will result in a smaller $Q$ since this decreases when approaching nadir. The ocean and thin sea ice have low $I$ and a high $Q$. As the sea ice gets thicker, the intensity increases and the polarization difference decreases. For the same incidence angle bias at higher thickness values $Q$ error will be smaller. The $I$ values for the two curves at the same SIT are nearly the same. The difference between these two curves is small compared to the difference to the SMOS 5.05 data version retrieval curve (Fig. 1 black).

Figure 3 shows the retrieved SIT using the daily mean method (left) presented in Sect. 3.1 and the retrained retrieval curve at nominal 45° incidence angle (center) based on the fitted TBs for 29 October 2010. Due to the requirement of the fitting procedure to have observations below 40° (Sect. 3.2), some grid cells in the central Arctic are not covered anymore. The decrease of the covered area surrounding the North Pole, relative to the old algorithm, is around 1° in latitude, corresponding

to approximately 1000 grid cells. This area is mostly covered by ice with thickness higher than 50 cm thus not being the focus of the retrieval. On the other hand for many ocean areas which formerly were excluded by the RFI filtering (grey in Fig. 3 left) now data is available, e.g. North Eastern Greenland. At the same time in the Hudson Bay area there is a 30% decrease in the covered surface due to failing the incidence angle criteria (Sect. 3.2) or the failure of the least square procedure to

converge to a solution. For 90% of the grid points the difference is less than 3 cm which is below the estimated retrieval error of 30% of SIT computed in Huntemann et al. (2014). The daily mean retrieval has a positive mean difference of 0.41 cm. The highest differences appear north of Alaska with values up to 10 cm (Fig. 3 right). This is a result of a biased distribution of the incidence angles, resulting in a large number of grid points having under 45° mean incidence angle. This decreases the polarization difference dragging the resulting SIT to higher values. Overall the RMSD for this day is 1.9 cm which is within

the expected 30% error margin of the retrieval.

Figure 4 (top) represents the mean difference (blue) and RMSD (red) of the SIT based on the 45° incidence angle fitted TBs relative to the 40-50° daily mean SIT calculated for the period of 1 October to 26 December 2010. To compute the mean difference and the RMSD we first divide the daily mean SIT into bins of 1 cm thickness, from 0 to 50 cm. To compute the mean difference for each 1 cm bin, we select all grid cells with thickness falling within that bin from the daily averaged SIT

and subtracting the thicknesses of the same grid cells obtained from the fitted TBs. The RMSD is also calculated between the two datasets for each 1 cm bin. Only grid cells that contain at most 50 cm are used. Also there must be a non-zero thickness in at least one of the two algorithms so that the high number of open water grid cells in both algorithms won't influence the statistics. Overall the SIT from the fitted TB is smaller than the SIT from the 40-50° incidence angle mean TB. For ice less than 40 cm thick the mean difference varies between 0 and −1 cm and then increases gradually up to -5 cm at 50 cm SIT. The green

curve shows the cumulative histogram for daily mean TB at each SIT. Approximately 52% of the ice thickness differences are below or equal to 3 cm in the daily averaged TB SIT. This can be explained by the coarse resolution of about 43 km of SMOS falsely generating thin sea ice at the ice edge due to TB contamination from either the ocean or the ice pack. In addition also coastal areas will spuriously generate thin sea ice due to spillover effects. Overall we can see that 95% of all data is below 40 cm while thickness values corresponding to 40 and 50 cm are contained in the remaining 5% of the data so that the region

of high mean difference is small. Figure 4 bottom shows the daily mean difference (blue) and RMSD (red) of the 45° fitted TBs SIT relative to the daily average TB SIT. Over the whole period the mean difference stays between 0 and −0.6 cm while the RMSD increases from  1.3 K to 2.5 K. The increase in RMSD can be explained by the freeze-up period which contains larger areas with intermediate thicknesses compared to the start and peak freeze-up periods which contain either ocean or over 50 cm SIT grid cells. The 45° fitted TBs SIT overall mean difference for the whole period for all thicknesses is −0.3 cm with

an RMSD of 2.02 cm.

## 4  Sea ice thickness retrieval using SMAP data

This section describes the adaptation of SMOS based SIT to SMAP TBs. Because SMOS observations have a variable incidence angle, they have to be computed at the fixed incidence angle of SMAP using the fitting function method described in Section 3.2.

In order to apply SMOS calibrated SIT retrieval to SMAP, first the TBs of both sensors have to be inter-calibrated (Sec. 4.1). In Section 4.2 the resulting inter-calibrated TBs are mixed and used for generating a combined SMOS/SMAP SIT dataset.

## 4.1 SMAP/SMOS inter-calibration

The first step is to retrain the SMOS retrieval as in Sect. 3.3 using the nominal incidence angle of 40°, which is the fixed incidence angle of SMAP. The resulting SIT retrieval curve is shown in red in Fig. 1. As expected, the lower incidence angle results in a lower $Q$, especially for thin ice and reduces the usable $Q$ range for the retrieval from 22-54 K to 17-43 K. Although the decrease of the dynamic range can increase the sensitivity of the retrieval to small changes in $Q$, the change is non-linear. At small thicknesses the decrease in dynamic range is large, 11 K at 0 cm, while the reduction of the dynamic range at 50 cm is approximately 5 K. The result is that the large change in dynamic range is affecting the low thicknesses which have low sensitivity to the change of $Q$.

A procedure to convert between SMOS and SMAP TBs over land was previously suggested in Lannoy et al. (2015). It uses a radiative transfer model and auxiliary data to account for atmospheric and galactic contributions for SMOS. For the interpolation of SMOS TBs to 40° incidence angle it fits a quadratic function to the angular dependent SMOS TBs.

In this study the procedure to convert from SMAP TBs to SMOS equivalent TBs is done through simple linear regression. For the procedure we use SMOS 40° measurements data and SMAP L1B TOA observations for the period between 1 October to 31 December 2015, which covers the first freeze-up in the Arctic observed by both sensors. All the data over 55°N is considered for intercalibration. In the first step, the SMAP data is gridded daily on the SMOS ISEA 4H9 grid (the native SMOS Level 1C data grid) using a Gaussian resampling with a cutoff distance from the grid cell center of 20 km and Full Width Half Maximum (FWHM) range of 40 km. Only grid cells located more than 100 km away from the coast are considered to minimize the land contamination. In the second step we determine the fit function parameters for the SMOS data on a daily basis and compute the 40° SMOS TBs for each grid cell. Figure 5 shows the scatter plots between the TBs of SMAP and SMOS 40° for horizontal (left) and vertical (right) polarization. For each polarization the magenta line shows the linear regression. We can distinguish two areas of high data point density at the two ends of the open water and thick sea ice clouds, respectively. Over open water at a TB of 80 K and 120 K for $TB_h$ and $TB_v$, respectively, SMOS has a positive mean difference of approximately 3.3 K and 5.2 K. At the high TBs representing the solid ice cover, the mean difference for SMOS decreases to 2.7 K and 3.3 K for $TB_h$ and $TB_v$, respectively. The bias of SMOS TBs in the 6.20 data version that is presented in Section 2 can be one of the sources for the difference between SMOS and SMAP TBs. The asymmetry between low TBs and high TBs can come from the high and low reflectivities of ocean and sea ice, respectively at L-band. Unlike SMAP, SMOS data does not include correction for galactic noise which can have a higher influence over water due to its high reflectivity. The reflectivity decreases over sea ice, resulting in galactic noise having a smaller impact on recorded values thus lower differences between corrected and uncorrected TBs. The overall RMSD of the two linear regressions is 2.7 K and 2.81 K for $TB_h$ and $TB_v$, respectively. The resulting linear regression parameters are presented in Table 2.

For this study, in order to use SMAP data for SIT retrieval, we adjust the SMAP TBs by a linear regression to 40° SMOS incidence angle TBs. A similar calibration of SMAP to SMOS TBs was presented previously in Huntemann et al. (2016). The

calibration was done through two separate linear regressions. The SMAP and SMOS 38-42° incidence angle data was daily averaged and compared to each other for the period 1 October to 31 December 2015 (just as is done here in Sect. 4.1). In the second step, since the SMOS SIT retrieval algorithm used in Huntemann et al. (2016) was developed for 40-50° daily averaged data another calibration is required. Using SMOS L1C data for the same period a linear regression is done between SMOS 40-50° and SMOS 38-42° daily averaged data.

There are two main differences between the Huntemann et al. (2016) and the current paper. The first difference is that here we use SMAP Level 1 B TOA data which does not include atmospheric correction instead of the surface TBs used in Huntemann et al. (2016). This is done to use comparable SMAP data to the SMOS TOA data that is used here. The second difference is that the SIT retrieval has been retrained to the fixed incidence angle of 40° and it is not necessary anymore to correlate SMAP TBs with the 40-50° SMOS averaged TBs. Instead we retrain the retrieval to work directly with 40° SMOS and SMAP TBs. Since the incidence angle difference between the SMAP data and the SMOS 40-50° data does not need to be corrected anymore, the calibration that is done in the current paper is necessary for (i) compensating for extraterrestrial contributions that are corrected in SMAP TBs and (ii) for the warm bias of the SMOS data. As a consequence, the transition from SMAP to SMOS TBs requires now just one linear regression compared to Huntemann et al. (2016). In this linear regression between the revised SMOS and the SMAP TBs (Sect. 4.1 and 4.2), the RMSD reduced by more than 1.3 K, approximately 30%, compared to Huntemann et al. (2016), indicating a better match of SMOS and SMAP based brightness temperature. This, in turn, ensures smaller differences between the retrieved SIT of both instruments and allows the combination of the two retrievals into a joint SIT product.

In Schmitt and Kaleschke (2018) a similar comparison is done to represent the differences between the SMOS and SMAP TB datasets. Compared with the inter-calibration done here, two years of data is used instead of three month, covering also the summer period over the Arctic Ocean. Since we consider that the algorithm presented here is valid just during the winter period, a calibration that covers summer months is not necessary. The RMSD between the SMOS and adjusted SMAP TBs in Schmitt and Kaleschke (2018) are between 1 to 3 K, which is in the same range of values presented in this paper, i.e. 2.7 and 2.81 K for $TB_h$ and $TB_v$, respectively.

For a daily SIT retrieval, based on horizontal and vertical SMAP TBs, the TBs first are adjusted to the SMOS TB using the linear regression parameters. Then they are gridded into a 12.5 km resolution polar stereographic grid using a Gaussian weighting for the distance with a cut-off from the grid cell center of 15 km and FWHM range of 40 km. For the period from 1 October to 31 December 2015, the difference in SITs between SMOS 40° incidence angle fitted TB retrieval and SMAP retrieval are small. Using grid cells containing SIT≤50 cm and at least one of the two retrivals having SIT>0 cm, the average difference for the SMOS SIT relative to the SMAP SIT is −0.2 cm with a RMSD of 2.39 cm.

For comparison, the bias and RMSD between SMOS and SMAP SIT found in Schmitt and Kaleschke (2018) are 1 cm and 7 cm, respectively, which is slightly larger than the results presented here. However, the time period considered in Schmitt and Kaleschke (2018) is different and the SIT retrieval is based on Tian-Kunze et al. (2014), thus having a different underlying principle.

## 4.2 SMOS/SMAP combined sea ice thickness retrieval

Because of the small differences between the retrievals from the two sensors, combined maps are produced using both of them. The daily mean horizontal and vertical TBs are computed separately for both sensors. For each grid point of the SMOS ISEA 4H9 grid we compute the daily SMOS TBs using the 40° fit (as in Sect. 3.3). Then the TBs are regridded to the NSIDC 12.5 km grid commonly used for sea ice maps. SMAP TB data is gridded directly to the NSIDC grid using a Gaussian resampling as was done in Sect. 4.1. The two resulting TB datasets are averaged. Finally the SIT retrieval for 40° incidence angle is applied. The result is a SIT map that has the benefit of using data from both sensors (e.g. Fig. 6 (left)) which has greater coverage, and is less affected by RFI. For the area north of 55.7°N the coverage in the mixed dataset increases by over 6% compared to the 40-50° daily mean TB retrieval. Also the combined TBs are more representative for a daily mean due to the 12 hours difference in the equator crossing time between the two sensors. The RMSD between the original 40° to 50° incidence angle daily mean retrieval from Sect. 3.1 and the new mixed sensor one is 2.05 cm for the 1 October to the 31 December 2015 period investigated, while the mean difference is -0.58 cm. This result means that the mixed sensor SIT is on average smaller than the SMOS daily averaged TB SIT. Figure 6 center shows the difference between SMOS 40-50° incidence angle averaged TBs SIT and the mixed data for the 24 October 2015. The greatest differences appear mostly in the transition area of 40 cm to over 50 cm. Taking into account just data points with maximum value of 50 cm and for at least one of the two datasets a value over 0 cm, 93% of the data has an absolute difference of at most 2 cm for the three months compared. Figure 6 (right panel) compares the retrieval done just with the SMOS 40° fitted TBs to the mixed data one. For this comparison, the average difference is below -0.1 cm and the RMSD is 1.37 cm for the complete three months period.

### 4.2.1 SMOS/SMAP combined sea ice thickness retrieval algorithm summary

To reach the final objective of the paper, combining TB data from both SMOS and SMAP sensors for a one day SIT retrieval several steps are required:

- SMOS L1C data is read and converted to the (H,V) reference frame (Sect. 2) and the data are limited to the region covered by the NSIDC polar stereographic grid

- for each SMOS grid cell the fit parameters for both H and V (Eq. 3) and corresponding uncertainties are derived (Sect. 3.2) and observations not covering 40° incidence angle are excluded.

- landmask is applied

- TBs at 40° are derived from the fit parameters (using the procedure from Sect. 3.3 and as applied in Sect. 4.2)

- the resulting TBs and uncertainties are gridded to the NSIDC polar stereographic 12.5 km grid

- SMAP L1B data is read and cropped to a minimum latitude of 55°N

- TOA TBs of SMAP are gridded to the NSIDC polar stereographic 12.5 km resolution grid (Sect. 4.1). TB uncertainties are an output of this step (Sect. 5.2)

- the gridded SMAP TBs are converted to SMOS equivalent TBs by linear regression (Sect. 4.1)

- for each NSIDC grid cell the SMOS and the converted SMAP TBs are averaged to obtain the combined TBs (Sect. 4.2)

- the uncertainties for the combined TB (for each polarization) are computed by error propagation from the uncertainties of $TB_h$ and $TB_v$ from SMOS and SMAP (Sect. 5.2)

- Polarization difference ($Q$) and Intensity ($I$) are calculated from the combined TBs ; the associated uncertainties are calculated by from the combined $TB_h$ and $TB_v$ uncertainties (Sect. 5.2)

- SIT is computed from each ($Q$,$I$) pair (Sect. 3.1) ; the uncertainties associated are computed at the same step using the results of the sensitivity study procedure discussed in Sect. 5

- additionally, after the gridding procedure for each sensor, SIT computation is done separately also for both SMOS and SMAP, using the same procedure presented above but by using the TBs and uncertainties of the specific sensor instead of the combined ones

## 5  Assessment of uncertainties

### 5.1  Sea Ice Concentration impact

The SIT retrieval used in this paper assumes 100% ice concentration. As a result, the retrieved SIT decreases if this condition is not fulfilled. We assume that TB over sea ice varies linearly with the change in sea ice concentration:

$$TBp(SIT,IC) = TBp_i(SIT) \cdot IC + TBp_w \cdot (1-IC) \tag{4}$$

where $p$ represents the polarization, $TBp_i$ and $TBp_w$ are the TBs of ice and water, respectively and $IC$ is the sea ice concentration.

For this study, as a first step, we first use $40°$ SMOS TBs from 11 October 2015 for retrieval. The resulting SIT will be considered the Ice Thickness (IT) for the assumption that we have a 100% ice concentration. In the second step we take the same TBs as input for the sea ice $TBp_i$, use fixed tie points for $TBp_w$ with 85 K and 125 K as values for the horizontal and vertical TBs, respectively. For each pair of SMOS TBs used in the first step we consider a range of sea ice concentrations (15, 30, 50, 70, 80 and 90%) for which we compute SIT using Eq. 4. The result is an IT value with its corresponding set of six SIC influenced SIT. As a last step, the IT data points are grouped in bins of 1 cm thickness. For each 1 cm bin of IT, we select its corresponding thicknesses from the second step and we averaged them for each SIC separately. Figure 7 shows how the retrieved SIT varies relative to the IT depending on the SIC. For a SIC of 90% at 10 cm the retrieved SIT is 8.5 cm, while at 50 cm is just 28 cm.

Current retrievals for SIC are influenced by thin sea ice. In Heygster et al. (2014), SIC algorithms have been tested for 100% sea ice concentration with thicknesses below 50 cm. All algorithms show less than 100% SIC for thicknesses below 30 cm. In

Ivanova et al. (2015) all SIC algorithms registered a decrease in SIC, up to 60% at 5 cm, and an overall bias of 5% for over 30 cm. An attempt to retrieve both SIC and SIT at the same time done in Kaleschke et al. (2013) showed a strong increase in noise for the SIT retrieval.

During the winter most of the Arctic is covered by SIC of 90% and higher (Andersen et al., 2007). For an assumed uncertainty of the sea ice concentration data of 4% (Ivanova et al., 2015) the error that could be introduced by a correction of SIT for high SIC is higher than that of the error introduced by the assumption of 100% sea ice concentration (Tian-Kunze et al., 2014). The uncertainty of SIC algorithms at high concentration and their covariation at thin thicknesses will cause high errors if a correction to SIT is applied using current SIC datasets. As a result full ice cover is assumed for the SIT retrieval.

### 5.2   Sea ice thickness uncertainties

In the SIT retrieval using $40°$ incidence angle TBs of the two sensors several factors contribute to the uncertainty: the radiometric accuracy of the observations, RFI contamination in the TB data, the uncertainty in the auxiliary data used for the training of the retrieval, the influence of the SIC on the TBs and the sub-daily variability of the TBs themselves.

Here we propose a method to quantify the uncertainty of the retrieval. We first compute the SIT in the $(Q, I)$ space using the $40°$ TBs trained retrieval (Fig. 8 (left)). The TBs that that will be used in a retrieval will more likely be found close to retrieval curve (Fig. 1 (red)) but there is variability, with data points, with values going above and below the curve. To cover also the less likely $(Q, I)$ pairs we chose to cover a large range of values for $Q$ and $I$, from 0 K to 80 K and from 80 K to 300 K, respectively. The resulting figure follows the training curve pattern, with an $I$ dominating the change in SIT below 20 cm thickness, while $Q$ becomes more important at higher thicknesses. The SIT over 51 cm is removed from the figure since we restrict maximum retrieved thicknesses to 50 cm. The one cm thickness over 50 cm is kept so that we can compute the derivative for 50 cm.

As a second step we compute the derivative as SIT as a function of $Q$ and $I$ seen in Fig. 8 center and right, respectively. For $Q$ values below the 20 cm line the change rate is below 0.25 cm per K due the thickness isolines being parallel with the $Q$ axis thus for the same value of the intensity, a large change in $Q$ will result in a similar thickness value. For thicknesses between 20 and 40 cm the change increases to 0.5 cm per k for $Q$ below 60 K. While for thicknesses over 40 cm the change rate of thickness with $Q$ quickly goes over one cm, especially in the area with $Q$ between 20 and 30 K where most of the data points will fall in. A similar patter appears also for $I$ with the difference that at thicknesses below 20 cm the change rate of SIT is higher than the one from $Q$ due to $I$ axis being perpendicular to the SIT isolines.The sensitivity of SIT relative to $Q$ and $I$ will be used to compute the uncertainty of the retrieval. For a given pair $(Q, I)$ and their associated uncertainties we compute the SIT and corresponding SIT uncertainties:

$$\sigma_{SIT} = \sqrt{\left(\frac{\partial SIT}{\partial Q}\right)^2 \cdot \sigma_Q^2 + \left(\frac{\partial SIT}{\partial I}\right)^2 \cdot \sigma_I^2 + 2 \cdot \left(\frac{\partial SIT}{\partial Q}\right) \cdot \left(\frac{\partial SIT}{\partial I}\right) \cdot \sigma_Q \cdot \sigma_I \cdot \rho_{QI}} \tag{5}$$

where $\sigma_Q$ and $\sigma_I$ represent the $Q$ and $I$ uncertainties derived through an error propagation method from the errors of $TB_h$ and $TB_v$, and $\rho_{QI}$ is the correlation between the $Q$ and $I$. The values of the SIT derivatives are taken from the second step of the method for each pair of $(Q, I)$.

For this study we do not take into account the radiometric accuracy of either sensor because they are small compared to the other errors, especially the TB variation during one day. For each SMOS observation at 40°incidence angle, the TB uncertainty is assumed to be the RMSD resulting from the fitting process presented in Sect. 3.2. During the fitting routine the RMSD is computed for each iteration and a 5 K threshold is used for eliminating outliers. Although this process is used to eliminate

potential RFI influences in the data, it will also reduce the variability that comes from observations of the same grid cell at different times of the day. For SMAP TBs a weighted standard deviation for each grid cell using all observations from one day is used as uncertainty. The weights are applied for each data point that is considered into calculating the TB for that grid cell and are computed using

$$w_i = \exp\left(-\frac{4 \cdot \log 2 \cdot d^2}{\text{FWHM}^2}\right) \tag{6}$$

where, $w_i$ is the weight, $d$ is the distance of the SMAP data point location to the center of the grid cell and FWHM is the Full Width Half Maximum beamwidth of SMAP with a value of 40 km. The correlation between the $Q$ and $I$ is -0.68 and -0.66 for SMOS and SMAP, respectively. The correlation was calculated for the period 1 October to 31 December 2015. It was computed for the whole three months over the whole Arctic using daily fitted TBs for SMOS and daily gaussian resampled TBs for SMAP for each grid cell.

Another source of error for the current retrieval is the uncertainty in the training data. For this study we included two parameters that could generate uncertainty in the creation of the retrieval curve and thus in the retrieval itself. The first parameter is the SIC. In the training data as presented in Huntemann et al. (2014) the SIC is assumed to be 100% although this cannot be ensured for the whole period covered. The initial freeze-up period, where thin sea ice can covary with SIC (as discussed here in Sect. 5.1), is allowed SIC between 0 and 100%, while later drops in SIC are removed. To take in account the uncertainty

in the SIC data used for the training, we take one day of TBs and corresponding SIT data and order it in 1 cm bins from 0 to 50 cm. Then we vary the SIC taken in account with ±5% standard deviation and compute the range of ice thickness that will derive from this, i.e., assuming 105% SIC and 95% using the linear mixing of open water contribution to TBs as discussed in Sect. 5.1. The result is shown in Fig. 9 (left). A 5% variation in the SIC for an assumed 100% SIC cover we obtain a polynomial increase in SIT error with increasing SIT, starting from nothing at 0 cm and reaching approximately 31 cm at 50 cm.

The second additional parameter used for estimating error in the retrieval curve comes from the CFDD daily variability in the estimation of training ice thickness using the model. While SMOS passes over a training area in the Arctic region, the recorded TBs are representative for that specific time of the over pass. Close to the poles a specific location can be covered multiple time by consecutive overpasses. For the generation of the retrieval curve, connecting the daily average temperature from NCEP with a localized in time daily averaged TB will create a bias between the retrieved thickness and actual SIT. The

variation in temperature, with lower temperatures increasing the ice generation rate, and it's non-linearity, with thinner ice growing faster for the same temperature than thicker sea ice, generates an uncertainty in the SIT computed for the retrieval curve. For quantifying this uncertainty we will select a fixed daily temperature of -25°C for which we compute the amount of thickness increase for 1 cm thickness as a starting point. This thickness will be considered the uncertainty of the SIT retrieval due to incorrect representation of the total sea ice increase in a day relative to the recorded TBs in the training areas. The result

is shown in Fig. 9 (right). At small thicknesses the error added by the CFDD daily variability is over 5 cm due to the greater exchange of heat between the ocean and the atmosphere, while it decreases exponentially towards 1 cm at the higher thickness. Also it can be seen that lower temperatures will increase the error due to greater exchange of heat between the ocean and the atmosphere.

To derive the final uncertainty for SIT, we use a simple error propagation method for the three uncertainty values that we want to include: uncertainty derived from the TBs and the associated retrieval, the uncertainty in the SIC training data and the uncertainty due to CFDD daily variability. Figure 10 shows as an example the scatter plot and moving average (red lines) of the SIT uncertainty (Eq. 5) for 24 October 2015 for SMOS (top) and SMAP (bottom). The restrictions imposed on the RMSD of the SMOS data have a clear impact on results. The TB uncertainties for SMOS in majority over 2 K lead at higher thicknesses

to high uncertainty. Because the SMAP data is still containing the full daily variability of observations, there will be grid cells with over 5 K uncertainty, but overall the median is around 1.2 K, in comparison with SMOS where the uncertainties are clustered around 4 K. Again, the smaller uncertainty of the SMOS data is only due to the TB fitting procedure, which removes outliers. Without that, for the raw data, the SMOS uncertainty would be similar or even larger than for SMAP. The CFDD daily variability uncertainty offers an offset of the SIT uncertainty relative to the zero line until approximately 20 cm. For both

sensors we can observe a rapid increase of the uncertainties beyond 20 cm SIT (Fig. 10) which can be explained by the high impact of SIC and the high sensitivity of the retrieval at values over 30 cm.

## 6    Comparison to ship based observations

Due to the nature of thin sea ice, in situ observations are extremely rare. Thin sea ice appears usually during the initial stages of the freeze-up period. Depending on the surface radiative energy fluxes and precipitation the sea ice growth may vary. From

the initial formation of sea ice to 50 cm thickness it may take less than one month. This can leave a short amount of time for in situ observations. In this section we will compare the SIT recorded from the R/V Sikuliaq during the period 5 October to 4 November 2015 in the Beaufort and Chukchi Seas with SIT data obtained from our combined SMOS/SMAP product. With more than 75% of the ship observations being of thin ice below 50 cm ice thickness, the dataset is well suited for comparison to the SMOS/SMAP product presented in this paper. The SIT and SIC data recorded by the ship was done mainly by hourly

visual ice observations using the ASPeCT protocol (Worby and Ackley, 2000). During the day, this allowed for an estimate of ice thickness in an approximate radius of 1 km, while during the night just in the ship vicinity covered by the floodlights.

We divide the ship data into separate days, and average the ice thicknesses within a 20 km radius from the center of each 12.5 km sized NSIDC grid cell. Figure 11 shows the comparison between the SMOS/SMAP product and the ship-based observations with the color indicating the ice concentration. The estimation of the ice area fraction was done using the ASI

ice concentration product from the University of Bremen (www.seaice.uni-bremen.de, Spreen et al. (2008)) resampled to the 12.5km grid. The points are well aligned around the one-to-one line even though with a high scatter. We eliminate grid cells which contain in the ship data thicknesses between 60 and 120 cm. With the remaining data we compute a linear regression of the two datasets which results in a slope of 0.71, an RMSD of 6.58 cm and a correlation coefficient of 0.58. Thus, SMOS/SMAP

is slightly overestimating the ice thickness compared to the ship observations. On the other hand, in this comparison, no SMOS/SMAP observations show higher ice thicknesses than 30 cm which may be caused by the reduced ice concentrations, e.g., for 90% SIC the retrieved SIT cannot be higher than 30 cm (see Fig. 7). We can see that there is high covariance between the SIC and the SIT with most low thicknesses appearing in areas with low SIC. The outliers at high SIT are probably caused by the local effects, e.g., small pieces of very thick ice close to the ship while in a larger area in order of 20 km radius SMOS/SMAP footprints thin ice is dominant. The fact that most of the area was covered by thin ice makes it quite likely that larger area averages yield thinner ice compared to the local observations.

The comparison of ship based with satellite based observations is problematic as the scale of the observations differ by a large amount. Satellite footprint sizes from SMOS and SMAP are in an order of 20 km radius while the observations based on the ASPeCT protocol are very local with 1 km radius. With a straight route of the ship based ice observations through a SMOS/SMAP satellite footprint, only about 6% of the area is covered. Therefore this comparison heavily relies on the assumption of consistency of ice conditions, i.e., high spatial autocorrelation of ice thickness. Taking these differences into account, the comparison actually shows a quite promising agreement between the two datasets.

In Section 4.2 we showed that the SIT difference between the old and the new combined retrieval are relatively small. Since this current combined retrieval is based on the empirical (Huntemann et al., 2014) retrieval and training data, using an adaptation to the SMAP incidence angle, changed the RFI filtering methods and combination of two sensors the comparisons and validation of the original product are still valid. For example, a validation study for SMOS SIT data, which also includes the Huntemann et al. (2014) dataset, was done in March 2014 in the Barents Sea (Kaleschke et al., 2016). Measurements included an airborne laser scanner and radiometer, and both airborne and ship-based electromagnetic induction (EM) systems. In that comparison the SMOS ice thickness data is too thin ($-20$ cm) compared to the airborne measurements, opposite to what is found here in the comparison to the ship-based data. A good correlation of approximately $0.7$, however, was found between the airborne measurements with the SMOS SIT product. On the other hand, no correlation was found between the ship-based EM observed thickness and the SMOS product, while in our comparison with the ship observations based on the ASPeCT protocol we find a significant correlation of $0.58$.

# 7 Conclusions

The existing retrieval for thickness of thin sea ice (Huntemann et al., 2014) from the L-band sensor SMOS (launched 2009) has been adapted to SMAP (launched 2015) by (i) modifying the SMOS retrieval to use $40°$ incidence angle instead of the average in the range $40°$ to $50°$, and (ii) establishing a linear regression between the SMOS and SMAP TBs at $40°$ incidence angle.

To derive the SMOS TB at $40°$ incidence angle required for the first step, an analytical function is fitted to the incidence angle dependent TBs. SMAP top of the atmosphere data and the SMOS data fitted to the same incidence angle yield a small TB RMSD between the two datasets for both polarizations of 2.7 K and 2.81 K for $TB_h$ and $TB_v$, respectively. This is an improvement compared to previous attempts (Huntemann et al., 2016) where the RMSD for both polarizations was over 4 K. Moreover the SMOS based ice thickness retrieval has been adjusted to the new SMOS data version 6.20. The new

algorithm contains a new RFI filtering routine exploiting the dependence of the TBs on the incidence angle. This method improved coverage of previously RFI affected areas. Although the TB datasets of the two sensors are processed differently, the overall resulting thicknesses are similar, with SMOS TBs having smaller variability at lower thicknesses due to the iterative observations removal operation. The comparison with in situ data shows a good agreement between the combined product and the ship observations.

Concluding, the benefit of SMAP for retrieval of thickness of thin sea ice is twofold: first, the combined product has a better spatial and temporal coverage that in future studies can allow insights even on a sub-daily scale. The overall increase in spatial coverage is 6%, although most of this is found in the lower latitudes where the existence of sea ice is minimal. Second, SIT can be retrieved from any of the two sensors alone with similar accuracy, making the production chain more stable in the case of malfunction of one of the two sensors. The algorithm and processing introduced in this paper can be seen as an extension of the method presented in Huntemann et al. (2014) because both methods yield similar values for the retrieved SIT. Therefore, the comparisons done in Huntemann et al. (2014) can be used as an additional assessment of the quality of the product presented in this paper.

Maps of thin ice thickness for the winter season in the Arctic and Antarctic are processed on a daily basis and made available under www.seaice.uni-bremen.de. In an era when the Arctic melting season and area of first year ice are increasing also the areas covered by thin ice are increasing. The new merged SMOS/SMAP ice thickness dataset is consistent with previous SMOS-only based ice thickness retrievals and will allow to extend the thin sea ice thickness record into the future.

*Data availability.* https://seaice.uni-bremen.de

*Competing interests.* No competing interests are present

*Acknowledgements.* We gratefully acknowledge the support from the Transregional Collaborative Research Center (TR 172). "ArctiC Amplification: Climate Relevant Atmospheric and SurfaCe Processes, and Feedback mechanisms (AC)[3]" funded by the German Research Foundation (DFG, Deutsche Forschungsgemeinschaft) and by the EU Horizon 2020 INTAROS (Integrated Arctic Observing System).

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

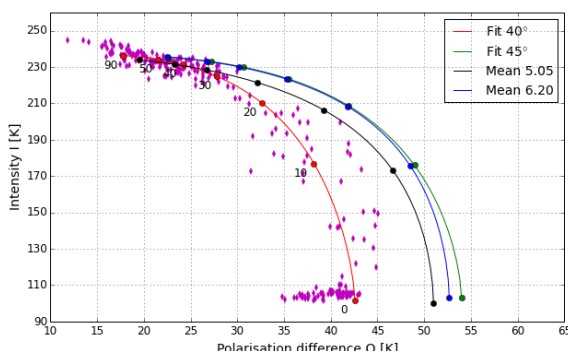

**Figure 1.** Sea Ice Thickness retrieval curves derived from SMOS data representing original algorithm (black), new data version (blue), 45° (green) and 40° (red) incidence angle fitted TBs. Dots represent data from the three training areas used for obtaining the 40° fit curve. Numbers under the curve represent the SIT in centimeters.

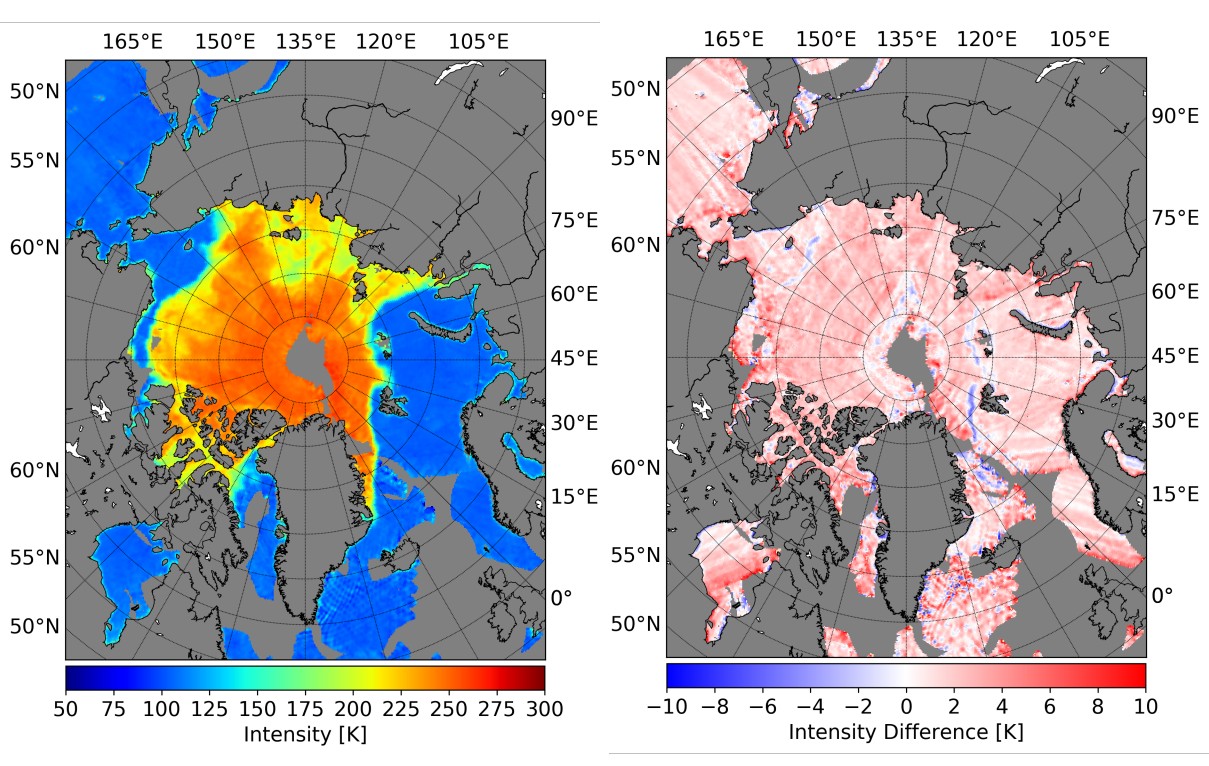

**Figure 2.** SMOS intensity for data version 6.20 data (left) for 29 October 2010 ; intensity difference (right) between the 6.20 and the 5.05 data versions.

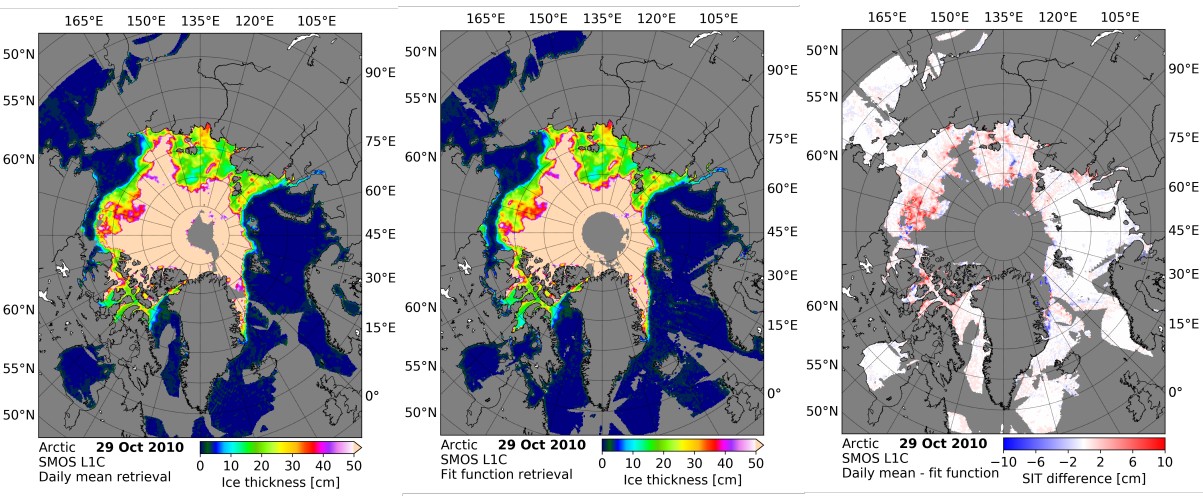

**Figure 3.** SMOS sea ice thickness retrieved on 29 October 2010 using 6.20 retrieval (left), retrieval using 45° incidence angle fitted TBs (central), and the difference between the two (right) with areas over 50 cm SIT not shown.

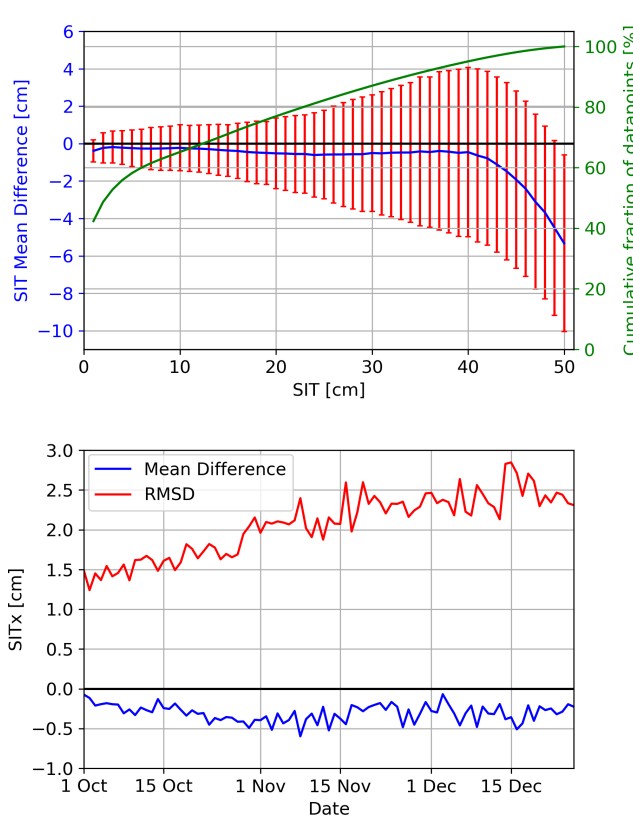

**Figure 4.** (Top) SIT mean difference (blue) calculated by substracting the SIT computed using 40-50° daily average from SIT using TBs fitted at 45° for the 1 Oct. to 26 Dec. 2010 period. Mean difference is computed relative to the daily average SIT in bins of 1 cm and its coresponding RMSD (red). Green curve represents the fraction from the total amount of data points for each thickness bin. Bottom figure shows the mean difference (blue) and RMSD (red) for each day separately.

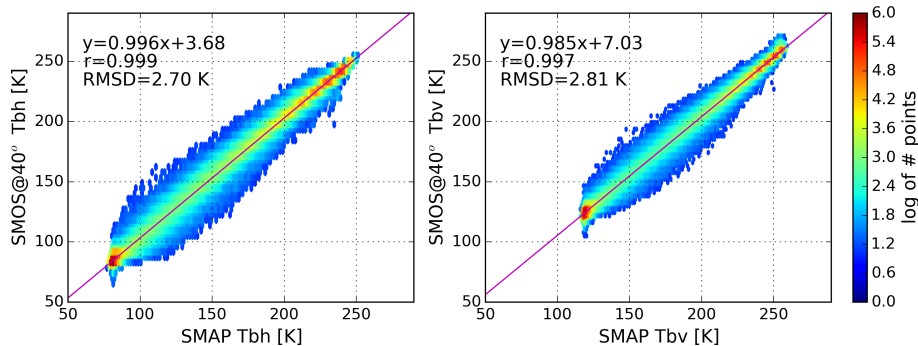

**Figure 5.** Logarithmic density plot of $TB_h$ (left) and $TB_v$ (right) data from SMAP and SMOS for the period 1 October to 31 December 2015. Magenta lines represent the linear regression between the two datasets.

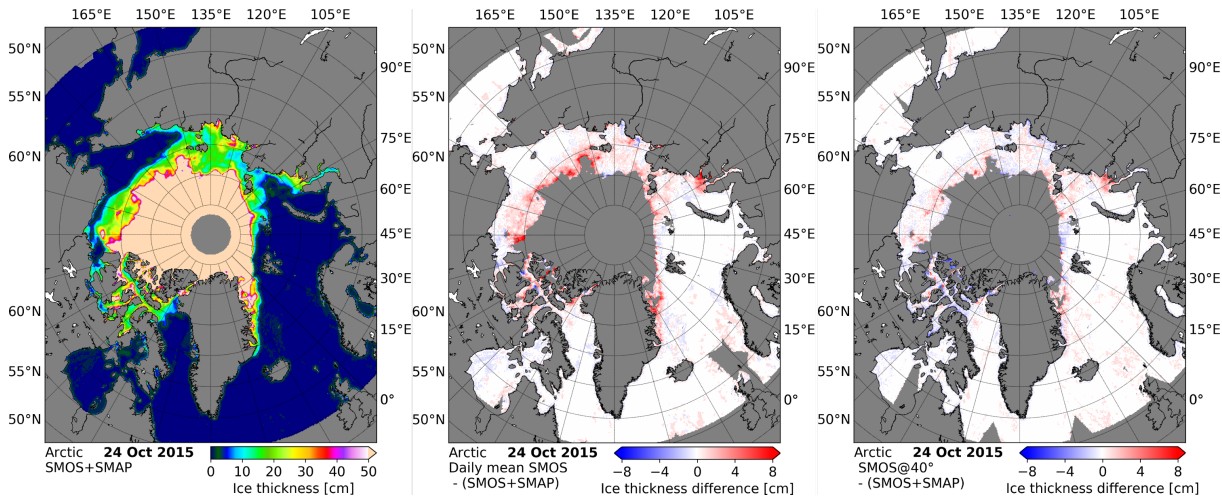

**Figure 6.** Sea Ice Thickness retrieved on 24 October 2015 for the joint SMOS+SMAP product (left), the SIT difference between the SMOS daily mean retrieval and the joint retrieval (center), and the SIT difference between SMOS fitted TBs at 40° incidence angle and the joint retrieval (right).

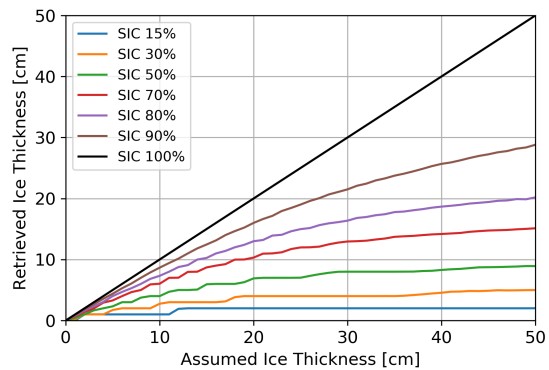

**Figure 7.** SIT retrieved as function of the assumed SIT under different SIC values.

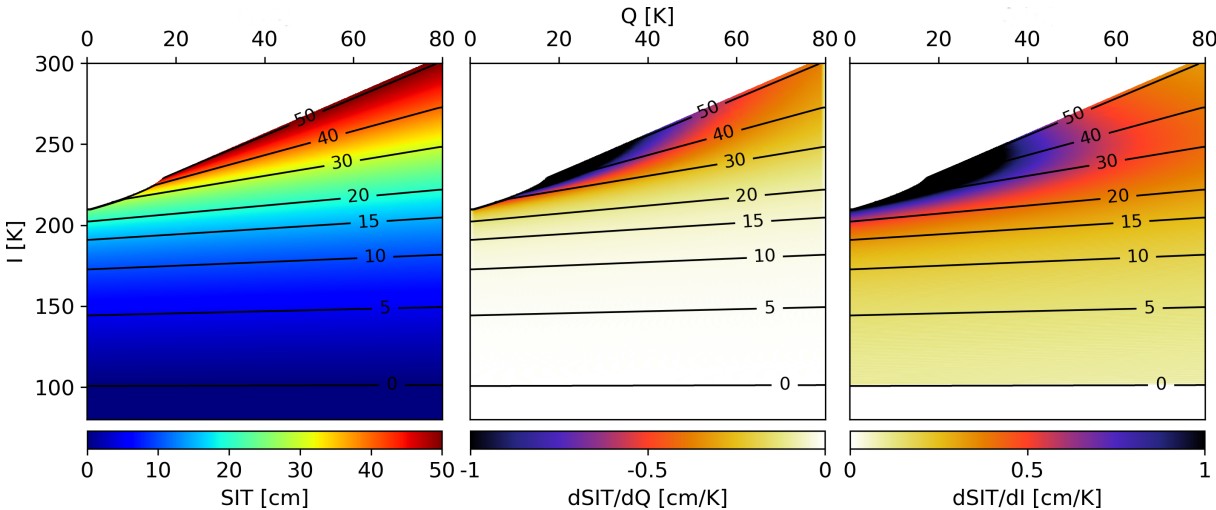

**Figure 8.** SIT (left) computed with the $40°$ TB algorithm (Fig. 1 red curve) represented in the space of $Q$ and $I$. Derivative of SIT as a function of $Q$ (center) and $I$ (right). The black lines in all three figures represent the isolines of the SIT derived from the left figure.

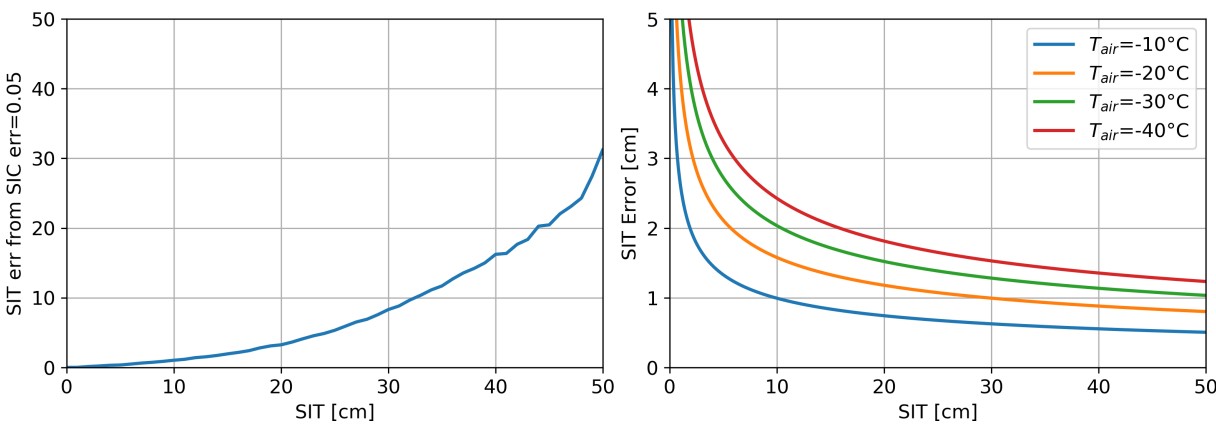

**Figure 9.** (Left) SIT error with change of SIT for a SIC uncertainty of 5%. (Right) SIT error as function of SIT due to CFDD daily variability calculated for various fixed 2 meters air temperatures.

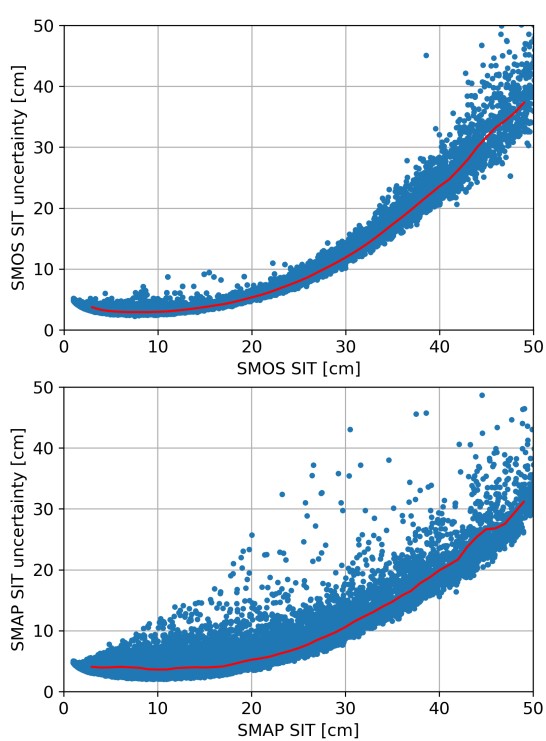

**Figure 10.** Scatter of SMOS (top) and SMAP (bottom) retrievals at 40°incidence angle for 24 Oct. 2015 in the Arctic and their respective uncertainties. The red line shows the rolling average of the uncertainty.

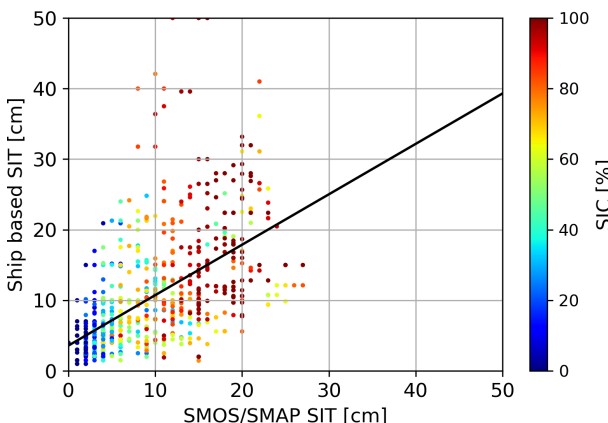

**Figure 11.** Comparision of ASPeCT based ice thicknesses observations by R/V Sikuliaq and the SMOS/SMAP retrieval. Ice concentration from the ASI product for the corresponding ice thickness observation is color-coded. The black line represents the linear regression of the two datasets.

**Table 1.** Sea ice thickness retrieval curve parameters for the original 5.05 data version training, 6.20 training, and the two fit curve parameters for 40° and 45° incidence angle

| Retrieval | Parameter | $a$ [K] | $b$ [K] | $c$ [cm] | $d$ |
|-----------|-----------|---------|---------|----------|-----|
| 5.05 | $I_{abc}$ | 234.1 | 100.2 | 12.7 | - |
| | $Q_{abcd}$ | 51.0 | 19.4 | 31.8 | 1.65 |
| 6.20 | $I_{abc}$ | 235.7 | 103.0 | 12.7 | - |
| | $Q_{abcd}$ | 52.7 | 22.3 | 33.2 | 1.60 |
| fit 40° | $I_{abc}$ | 236.4 | 101.5 | 12.2 | - |
| | $Q_{abcd}$ | 42.6 | 17.3 | 32.9 | 1.39 |
| fit 45° | $I_{abc}$ | 235.4 | 103.3 | 12.5 | - |
| | $Q_{abcd}$ | 54.0 | 22.2 | 33.0 | 1.47 |

**Table 2.** Parameters for linear regression between SMOS and SMAP TBs

| Polarization | Slope | Intercept [K] | RMSD [K] | r |
|:---:|:---:|:---:|:---:|:---:|
| *H* | 0.996 | 3.68 | 2.70 | 0.999 |
| *V* | 0.985 | 7.03 | 2.81 | 0.997 |