# Peer review of "Combined SMAP/SMOS Thin Sea Ice Thickness Retrieval"

_The Cryosphere, 2017_

## Referee Comment (RC1) · Anonymous Referee #1 · 29 Dec 2017

The paper entitled 'Combined SMAP/SMOS thin sea ice thickness retrieval' by Patilea et al. proposes a combined sea ice thickness (SIT) retrieval using SMOS and SMAP brightness temperature. First, authors present an algorithm to retrieve SIT from improved version of SMOS TB. Secondly, a SMAP calibration at 40 deg with SMOS is proposed. Finally, a combined SMOS+SMAP SIT retrieval is attempted. Although, the combined use of SMOS and SMAP circumvents several issues and has a lot of potential, more work needs to be done before the manuscript can be recommended for publication.

**General comments:**

Full validation, as mentioned by the authors on the last line of the paper, needs to be considered in this paper itself. Without validation, it is difficult to assess the performance of the proposed algorithm (s) – one from new version of SMOS TB, second from combined SMOS+SMAP.

The other major issue is the differentiation between contribution of TB from sea ice concentration (SIC) and thin sea ice. It is not clear from the paper, whether the authors assumed 100% SIC or above 80% or all 0-100%? If SIC less than about 80% is used, then thin ice retrieval is incorrect. A SIC map would be very useful. These point should be discussed in the paper.

Unfortunately, the paper does not show the validation part of this work, so it makes it very difficult to assess the retrieved SIT in the areas where SIC is below 80%. It is likely that the thin ice area shown in Figure 5 may be, in fact, marginal ice zone with SIC less than 80%.

I miss an error analysis, maybe you could cite the error budget from Huntemann.2014.

In Figure1, it would be easier to read if you show sit of the same date as shown in Figure 5 SIT retrieval, i.e. 29 Oct. 2010 OR 11 Oct. 2015.

There are spelling errors throughout the paper.

**Specific comments:**

P1.L5: 'SMOS data covers a large incidence angle range whereas SMAP observes at a fixed 40◦ incidence angle which makes thin sea ice thickness retrieval more stable as incidence angle effects do not have to be taken into account'. I do not agree that the incidence angle variation of SMOS brings instability on the measurements. Explain it better to convince the reader or remove the sentence. SMAP is more stable because it is a real aperture antenna, better RFI control system, narrower swath, etc...

P1.L20: The radar system of SMAP failed few months after launched. You should specify this, is you talk about the radar system.

P2.L12: 'Within a snapshot just one or two of the Stokes parameters are measured at the same time'. Please explain this properly, this sentence has no sense. Explain when one or two are measured?

P2. L23: 'SMAP started providing data starting in April 2015.' Re-phrase.

P.2.L8. It is not clear to me which is the RFI filtering method used in this paper, the one in Huntemann et al 2014, or 2015?

p3.L25: You should explain that a resolution of 12.5km is a subsampling grid, this is not the real SMOS resolution. In fact, in my opinion, it has not sense to go to 12.5Km, only with SMOS data, you should use the grid of 25km. In case you use other data with thinner resolution combined with SMOS then you could go to 12.5km, if not this resolution is completely false. You should explain this clearly in the paper.

Eq. 2: It is very dangerous to use in the equation of Q the exponential of a number which is raised to d (Q=(a-b).exp(-x/c)^d)+d), it makes the model inestable. But I imagine this comes from Huntemman 2014 method...

P3.L28. Please explain – spillover effects. Or Re-word.

P4.L9-10. 'Also…version.' Please re-write the sentence. It's unclear.

P4.L22: 'Moreover,…' , here you are talking about your method to remove RFI and averaging the TB, not on the SMOS acquisition method. This should be specified, if not, it seems this is a problem of SMOS, since the previous sentence you talk about the SMOS  acquisition method. This sentence starting with 'Moreover,' I understand is the basis to explore the new method of fitting a curve, so please explain this clearly.

P5.Eq 3: Are the values of a, b, c, d the ones in table 1? This is not cited here. And are bh and bv from the equation different? I don't see the values of H and V pol in table 1. Moreover, bh and bv, they should be equal, to permit to have Tbh=TBV at Theta=0 (nadir).

P5.L23. Figure 4 is mentioned in the text before Figure 3. Please correct it.

P6. L1: You talk about 'estimated retrieval error of 30% of SIT'. This is not commented before, from where do you have this value? Please add reference. Which method has this retrieval error? The daily of the 45º? Please discus better the error budgets as well.

Figures:

Figure 1 and 4: Increase font size.

Figure 5: Please also show the SIT map derived using SMAP only, and SMOS only for the same date alongside SMOS+SMAP SIT map. This will make it easier for visual comparison.

References:

- Please add webpage from where to find the documents used in the references (Indra Systems and SMOS Calibration…). I am sure they are public.

- I think the references should order it for name of first author and year. So reference 3 and 4 should be switch.

---

## Referee Comment (RC2) · Anonymous Referee #2 · 9 Jan 2018

The paper by Paţilea et al. deals with retrieving thin sea ice thickness (SIT) from L-band radiometry using brightness temperatures (TB) from two different satellite missions: SMOS and SMAP. The study mainly deals with three things: 1) An existing SIT retrieval is applied to a newer data version of SMOS. 2) A slightly different approach to fit TBs in the SMOS retrieval is applied (using a TB fit to a specific incidence angle, e.g. 45°, instead of using the average TB from measurements between 40 and 50°). 3) SMAP TBs are "converted" to be used in the SIT retrieval, which was originally set up for SMOS, and combined SMOS and SMAP SIT maps are produced.
The produced maps are compared with each other but not compared with independent SIT data.

Unfortunately, the used methods are not described precisely enough (see, for example, comments 6), 10), 11), 12) below). Furthermore, the Level 1B SMAP data is claimed to be "top of the atmosphere" TBs, i.e. atmospheric effects are not taken into account. However, according to the Algorithm Theoretical Basis Document (ATBD): "SMAP Calibrated, Time-Ordered Brightness Temperatures L1B_TB Data Product" [1]  and De Lannoy et al. (2015) [2], for example, SMAP Level 1B TB data are "corrected for atmospheric effects". As far as I know, this is also one of the reasons why SMOS (Level 1C) TBs (which are not corrected for atmospheric effects) and SMAP (Level 1B) TBs differ. Thus, it is not clear how the authors dealt with this. In general, I think the differences between SMOS and SMAP should be discussed and presented in more detail.

Another issue is that there is a study by Huntemann et al. (2016) [3] that deals with using SMOS and SMAP for ice thickness retrieval and shows a resulting map (it is mainly with the same authors). The paper ([3]) gives the difference between SMAP and SMOS retrieved SIT for Oct to Dec 2015 and contains a SMAP SIT map and the SIT difference between SMAP and SMOS for 7 Oct 2015, while the paper presented here compares the combined SMOS and SMAP SIT with SMOS SIT and shows a combined SMOS and SMAP SIT map and how it differs from SMOS SIT for 11 Oct 2015 (Fig. 5). As these two studies seem to deal with very similar things, I think it is very important to clearly state the difference and the "added value" of the study presented here. My main concern is that there is already a paper ([3]) that shows how SIT retrieved from SMAP compares with SIT retrieved from SMOS (for the TB polarization difference and intensity approach) for Oct to Dec 2015. The new thing in the study presented here seems to be that SMOS and SMAP SIT are not compared but instead averaged to form combined SIT maps, and these have been compared to SMOS-only maps for Oct to Dec 2015 (I think this is the period, see comment 19)). However, as it is not clear how the SMAP SIT or the combined SMOS and SMAP SIT compare with independent SIT data, just combining the two data sets, as done here, does not necessarily bring new information.

The manuscript contains several spelling errors that could have been avoided by simply using a spell checker (thus I will not point them out). Furthermore, the style of writing could be improved and the usage of commas should be revised. Not all statements are supported by references (or a reference is given that I cannot find/access), see specific comments below.

**Specific comments:**

1) Introduction:
- I think the introduction should be improved in that the work presented here is put more into a scientific context. Is this the first time SMOS and SMAP are merged? What are the "challenges"? Why not elaborate more on the differences between SMOS and SMAP? This is not mentioned at all in the introduction (only shortly in the abstract but that should be independent). In general, I think that the introduction (especially starting from p. 1, l. 22) may be hard to understand for those who are not very familiar with the SMOS SIT retrieval and the SMAP mission.

- No references given for statements made in the first lines of the introduction (p. 1, l. 10-13) and for "*The atmosphere has little influence on the radiation at L-band as both absorption and scattering are small.*" (p. 1, l. 16-17).

2) Several sentences start with "*It*" or "*This*", which can make reading hard because it is not always clear what "it" or "this" refers to (especially if referring to something mentioned in the last sentence or even earlier...).
Example 1 (four sentences in a row): "*and it will be used... This is combined ... It is used for... This is also...*" (p. 1, l. 24 - p. 2, l. 1)
Example 2: "*It adds better ... At the same time it ...*" (p. 2, l. 17-18)

3) p. 2, l. 9-15, including eq. (1): Maybe this section is not needed because this is not the focus of the study and seems to be the same approach as in Huntemann et al., 2014?

4) Statements on p. 2, l. 17 ("*It adds better RFI flagging...*") and l. 20-21 ("*The new data version also reduces...* ") are not supported by references. Is "SMOS Calibration team and Expert Support Laboratory Level 1, 2015" the reference for these? I cannot find or access this. Please provide a web link or other hints how to find this. Also the "Indra Sistemas S.A.: SMOS Level 1 and Auxiliary Data Products Specifications, Product Document, Madrid, 2015" reference may contain more information to make it more easy to find. References for statements in p. 2, l. 29 - p. 3, l. 3?

5) p. 2, l. 18-19: "*... it introduces a warm bias in the brightness temperatures of aproximately 1.4 K relative to the previous version 5.05 over ocean...*" -> Is 1.4 K warmer brightness temperatures over the ocean more/less realistic? Or just a trade off to get the SMOS brightness temperatures over land (and Antarctica) closer to modeled and measured values? But for this study this is not important anyways because no changes were observed for the high latitudes? This is somewhat confusing...

6) p. 2, l. 29 - p. 3, l. 5 two RFI filters are described. However, it is not mentioned which RFI filter approach is used in the study presented here. Later (p. 4, l. 28-30) you describe a third RFI filter method. Were different RFI filters used for different parts of this study?

7) p. 3, l. 12-14: "*retrieval based on a fitting function ... through the full incidence angle range. This will provide more stable brightness temperatures and is necessary for a consistently combined SMOS and SMAP ice thickness retrieval.*" -> What is the "*full incidence angle range*"? Why is this "more stable"? More stable than method by Huntemann et al., 2014? Is that a result of the work presented here? Or has that been shown somewhere else (where?)? Please clarify. Why is fitting through the full incidence angle range "*necessary for a consistently combined SMOS and SMAP*" retrieval if SMAP measures only at one incidence angle?

8) p. 4, l. 6: "*From a total of 5.1 million data points,...*" -> Where does this number come from? All Arctic data points from a certain time period?

9) p. 4, l. 19-22: Maybe a reference to a paper that describes the SMOS snapshot geometry would be useful here.

10) Where do the equations in eq. (3) come from? How is C "*determined by averaging the sum of the polarizations for each observation*" (p. 4, l. 33-34)? After all fit parameters ah,av,bh,bv,dv have been determined first (are they determined for each observation? for each incidence angle range? as generally valid values?) ?

11)  p. 5, l. 9-10: "*only grid cells with the incidence angle range of observations that covers the wanted angle, e.g. 45◦, are used for the retrieval*" -> What is the exact criterion here? (e.g. "only

grid cells that contain at least $N$ observations with incidence angles $\theta\pm\Delta\theta$ around the considered incidence angle $\theta$")

12) p. 5, l. 27-30:
-"*On the other hand for many ocean areas which formerly were excluded by the RFI filtering (grey in Fig. 4 left) now data is available, e.g. around Iceland, Eastern Greenland and Vladivostok.*" -> I don't know which TB measurements are used for the "fitted 45° brightness temperature" method... The text says that you need "*at least one observation under 40°*" and that you use "*only grid cells with the incidence angle range of observations that covers ... 45°*" (see also comment above). If the latter means, for example, that you use incidence angles between 40 and 50° PLUS at least one measurement with incidence angle <40° (which adds at least one more possibility to encounter an RFI-contaminated observation), how can this method be less influenced by RFI than the algorithm that uses the daily mean from measurements between 40 and 50°? Is it because of different RFI filters? (see also comment 6)) Is there a minimum number of measurements that has to be available in order to perform the SIT retrieval?
-"*At the same time in the area of the Hudson Bay there is a 30% decrease in the area covered due to the high uncertainty of the fit.*" -> How does the "*high uncertainty of the fit*" influence the number of grid cells the retrieval is performed on? Is there a criterion for cases in which the retrieval is terminated / not performed? Is that given anywhere in the manuscript?
- Is the same number of SMOS measurements used for both retrieval approaches here? SMOS data can be quite "scattered", the number of data points used/averaged in the retrieval can have an impact on the results...

13) In the manuscript text, Fig. 4 is mentioned before Fig. 3.

14) p. 6, l. 1: "*estimated retrieval error of 30% of SIT.*" -> Where does this number come from?

15) p. 6, l. 5: "*within the error margin of the retrieval*" -> Which error margin? From where? (the 30% given in l. 1?)

16) Do the RMSD and bias given on p. 6, l. 2-5 refer to the comparison of the two retrieval approaches for just the one day (29 Oct 2010) shown in Fig. 3 and 4? I wouldn't consider this representative...

17) p. 6, l. 16-17: You mention that Lannoy et al. (2015) take into account atmospheric contributions for SMOS "*to convert between SMOS and SMAP TBs*". You didn't, did you? (earlier (p. 2, l. 26) you claimed that SMAP TBs are "*top of atmosphere*"). Please write clearly what YOU have done to bring SMOS and SMAP TBs together and how and why your approach differs from other approaches etc.

18) p. 7, l. 1-2: "*For 11 October 2015 (not shown) the differences ... between SMOS fitted TBs retrieval and SMAP retrieval are small.*" Why do you pick one day to state this, and then not even show it? What about the other days? What is "SMOS fitted TBs retrieval" here? The one fitted to 40° incidence angle?

19) p. 7, l. 13-14: "*The RMSD between the original 40◦ to 50◦ incidence angle daily mean retrieval from Sect. 3.1 and the new mixed sensor one is 2.14 cm for the three months period investigated, while the bias is -0.63 cm...*"
-What is the "*three months period investigated*" here? The same as was used for calibration earlier (Oct to Dec 2015)?
-It would be interesting to know the difference between SIT from SMOS fitted to 40° and SMOS+SMAP; otherwise we cannot distinguish whether the difference comes mainly from the

different approaches for the SMOS retrieval (fitted to 40° or mean between 40 and 50°) or the different sensors (SMOS or SMOS+SMAP). Indeed, the values from the comparison in 3.3 are very similar (bias 0.69cm and RMSD 2.2cm), although a) I am not sure in which direction the negative sign in "-0.63 cm" points here, and b) the results in 3.3 seemed to be for a comparison of one day only.

20) In the conclusions, there seems to be a "new" result: "*Using SMAP data and the SMOS data fitted to the same incidence angle to calibrate the SMAP TBs to those of SMOS has improved the TB RMSD between the two datasets for both polarizations with 2.7 K and 2.81 K for TBh and TBv, respectively.*" (p.7, l. 22-24) -> This was not mentioned before. Do you mean improved to or by 2.7K? The sentence is hard to understand, also because the reference to the "improvement" comes only in the next sentence. It says "*This is an improvement from previous attempts (Huntemann et al., 2016) where the the RMSD for both polarizations was over 4 K showing that using fixed incidence angles for SMOS data increases the accuracy.*" The phrasing could be clearer (who concludes this?). And I don't agree... 1) This shows only that SMOS and SMAP TBs as compared here and in Huntemann et al. 2016 agree better if a fixed incidence angle is used. And as I don't know how you dealt with the atmospheric corrections for SMOS, I cannot say whether it is a good thing that the SMOS TBs you used here (not corrected for atmosphere? corrected differently than SMAP?) and the SMAP TBs (atmosphere corrected) are closer to each other or not... 2) Averaging SMOS TBs differently does not "*increase the accuracy*" of SMOS data, but these SMOS data might be more suitable for comparison with SMAP.

21) What about ice concentration? Has big impact on L-band TBs but is not mentioned at all.

22) The numbers given in Tab. 2 for the "*original 5.05 data version*" (SMOS) do not correspond to the numbers given in the literature for this SIT retrieval with SMOS (Huntemann et al., 2014 & Huntemann et al., 2016). (a and b interchanged for I_abc; different a, c, d for Q_abc).

23) Fig. 1 in the current print version is quite small, especially the label font size.

**Further comments:**

p. 1, l. 4-5: "... *SMAP observes at a fixed 40◦ incidence angle which makes thin sea ice thickness retrieval more stable as incidence angle effects do not have to be taken into account.*" -> Why would this be "*more stable*"? Isn't it mainly a restriction to not have any measurements at incidence angles other than 40°? For example, wouldn't a retrieval using only SMOS measurements at 40° be as stable?

p. 1, l. 20: Very abrupt and sudden transition from describing SMOS to introducing SMAP.

p. 1, l. 22-23: "... *by calibrating the SMAP brightness temperatures (TBs) to those of SMOS*" -> Why do SMAP measurements have to be calibrated to SMOS measurements? Here you could mention whether/how they differ. I guess this has also to do with SMOS brightness temperatures having been compared (or "validated") with other sources (in Huntemann et al., 2014, for example).

p. 2, l. 12: "*the the*"

p. 2, l. 16: "... *version 6.20 has been available since 5 May 2015 operationally...*" -> version 6.20 has been operationally available since 5 May 2015

p. 2, l. 24: "*a equator*" -> an equator

p. 2, l. 24: "*...data is used, which contain...*" -> "...data is used, which containS..." or "...data ARE used, which contain..." (but then "data" has to be used as a plural word throughout the paper)

p. 2, l. 27: "*36x47 km*" -> 36 km x 47 km

p. 3, l. 20: "*retrieval curve*" singular vs. two equations

p. 4, l. 7: "i*t increases*" -> they increase?

p. 4, l.10: "*This allows to estimate*" -> I think this is grammatically incorrect (while "This allows us to estimate..." should be ok I guess)

p. 4, l. 13-14: Remove "*given above*"?

p. 4, l. 29: "*observation*" -> observations

p. 7, l. 17-18: I recommend to combine these two lines to one because they are very repetitive.

p. 7, l. 19: "*rage"* -> range

p. 7, l. 25: "*the the"*

**References:**

[1] Algorithm Theoretical Basis Document (ATBD): "SMAP Calibrated, Time-Ordered Brightness Temperatures L1B_TB Data Product", J. Piepmeier, P. Mohammed, G. De Amici, E. Kim, J. Peng, and C. Ruf (2014), e.g. Tab. 1, p. 11, file accessed at https://www.google.de/url?sa=t&rct=j&q=&esrc=s&source=web&cd=3&ved=0ahUKEwiF2sqb3MDYAhVBa1AKHbJKAfIQFgg2MAI&url=https%3A%2F%2Fsmap.jpl.nasa.gov%2Fsystem%2Finternal_resources%2Fdetails%2Foriginal%2F278_L1B_TB_RevA_web.pdf&usg=AOvVaw34MP0zjMXDqHm83OB1qAog

[2] De Lannoy, G. J., Reichle, R. H., Peng, J., Kerr, Y., Castro, R., Kim, E. J., & Liu, Q. (2015). Converting between SMOS and SMAP level-1 brightness temperature observations over nonfrozen land. *IEEE Geoscience and Remote Sensing Letters, 12*(9), 1908-1912.

[3] Huntemann, M., Patilea, C., and Heygster, G. (2016): Thickness of thin sea ice retrieved from SMOS and SMAP, in: Proceedings of 2016 IEEE International Geoscience and Remote Sensing Symposium (IGARSS), pp. 5248–5251.

---

## Author Comment (AC1) · 2 Mar 2018

These are our answers to the two referees. We want to thank both of them for their comments and suggestion which helped to improve the manuscript. We hope that our explanations address the issues raised by the reviewers and will make the manuscript more clear.

We will thoroughly revise the manuscript to address the reviewers comments. In particular we have added a section discussing the uncertainties and sea ice concentration influence of the retrieval. Five more figures showing a) SIT bias and RMSD (daily variability and for bins of thickness) between daily averaged and 45 deg fit SMOS TBs - as a replacement for Fig. 3, b) Density plots with linear regression between SMOS 40 deg and SMAP data c) SIT sensitivity to change in horizontal and vertical brightness temperature ; d)SMOS 40 deg and SMAP SIT vs SIT uncertainty for a day 3) SIT retrieved as function of an assumed SIT under different sea ice concentration values. To Figure 5 we added a third map with SMOS at 40 deg SIT - SMOS+SMAP.

Answer to Anonymous Referee 1:

The paper entitled 'Combined SMAP/SMOS thin sea ice thickness retrieval' by Patilea et al. proposes a combined sea ice thickness (SIT) retrieval using SMOS and SMAP brightness temperature. First, authors present an algorithm to retrieve SIT from improved version of SMOS TB. Secondly, a SMAP calibration at 40 deg with SMOS is proposed. Finally, a combined SMOS+SMAP SIT retrieval is attempted. Although, the combined use of SMOS and SMAP circumvents several issues and has a lot of potential, more work needs to be done before the manuscript can be recommended for publication.

**General comments:**

Full validation, as mentioned by the authors on the last line of the paper, needs to be considered in this paper itself. Without validation, it is difficult to assess the performance of the proposed algorithm (s) – one from new version of SMOS TB, second from combined SMOS+SMAP.

The other major issue is the differentiation between contribution of TB from sea ice concentration (SIC) and thin sea ice. It is not clear from the paper, whether the authors assumed 100% SIC or above 80% or all 0-100%? If SIC less than about 80% is used, then thin ice retrieval is incorrect. A SIC map would be very useful. These point should be discussed in the paper.

As in Huntemann (2014) the same three areas (in the Kara and Barents Seas) over the same period were selected for the training of the SIT retrieval curve. The assumption is that SIC is 100%. For the training, SIC between 0 and 100 was allowed for the initial freeze-up to not miss the formation of very thin sea ice due to the SIC retrievals. Section 5.1 was included to discuss the SIT uncertainty  that is introduced by SIC uncertainty.

Unfortunately, the paper does not show the validation part of this work, so it makes it very difficult to assess the retrieved SIT in the areas where SIC is below 80%. It is likely that the thin ice area shown in Figure 5 may be, in fact, marginal ice zone with SIC less than 80%.

I miss an error analysis, maybe you could cite the error budget from Huntemann.2014.

The retrieval error of 30% that we refer to in the manuscript is taken from the Huntemann et al., (2014). We have added it as a reference at the necessary locations in the manuscript. You are right that for the MIZ with SIC less than 80% the heterogeneity within the SMOS and SMAP footprints will cause larger uncertainties and the derived ice thickness will include the open water areas. This is a problem of all SMOS thin ice thickness retrieval available today. To resolve that problem is not the purpose of this paper.

In Figure 1, it would be easier to read if you show sit of the same date as shown in Figure 5 SIT retrieval, i.e. 29 Oct. 2010 OR 11 Oct. 2015.

The 2010 date for Figure 1 was chosen because of two reasons:
1. it's in the freeze-up period of the 2010 which was used for the training of the SIT retrieval curve
2. it shows the difference in intensity for the same day between the two SMOS Level 1C data versions 5.05 and 6.20. The Huntemann paper has used 5.05 data for training the algorithm, data version that was discontinued after the release of 6.20. Thus the overlap available for comparison between the two datasets is pre-2015 freeze-up.

Since the new data version introduced some biases in the TBs as mentioned in P2L16-21 and the Huntemann algorithm together with the new data version was used later as a reference as mentioned in P4L16-18 we compared the Huntemann algorithm using the two data versions as shown in Section 3.1
As a result Figure 1 can't be redone in the freeze-up period of 2015 due to the lack of 5.05 data, while Figure 5 can't be moved to 2010 because it is before the launch of SMAP thus we cannot adapt the two figures to show the same date.

There are spelling errors throughout the paper.

We carefully reread the paper and corrected many spelling errors to the best of our knowledge.

**Specific comments:**

P1.L5: 'SMOS data covers a large incidence angle range whereas SMAP observes at a fixed 40∘ incidence angle which makes thin sea ice thickness retrieval more stable as incidence angle effects do not have to be taken into account'. I do not agree that the incidence angle variation of SMOS brings instability on the measurements. Explain it better to convince the reader or remove the sentence. SMAP is more stable because it is a real aperture antenna, better RFI control system, narrower swath, etc...

We considered mainly two factors for this affirmation:
1. depending on the position in the swath (closer to the center or to the edge) for a pixel, the distribution of the incidence angles recorded during one overpass will be different due to the nature of SMOS snapshot

2. the RFI filter used in Huntemann et al. removes a snapshot that contained even one data point >= 300 K. If I take as an example a source on land that is active during an overpass from land to ocean, this will result in the elimination of data points of higher incidence angle (if we consider an ocean pixel along the trackline) while after the source is cleared just the remaining incidence angle data points will be used for the mean.

Taking in account the previous two points if we consider the retrieval using the 40-50 deg mean brightness temperature, in some cases the resulting mean incidence angle for a pixel can be be at the ends of the this range. Due to the divergence of the horizontal and vertical brightness temperatures with increasing incidence angle, the resulting polarization difference can change significantly. Since the divergence is asynchronous (with horizontal Tbs decreasing faster than the increase of the vertical polarization Tbs) especially at lower brightness temperatures also the average intensity can change.
Since the retrieval doesn't take into account the actual average incidence angle it introduces additional uncertainty.
The sentence will be change to: "SMOS data covers a large incidence angle range whereas SMAP observes at a fixed 40 deg incidence angle which makes thin sea ice thickness retrieval more consistent as incidence angle effects do not have to be taken into account".

P1.L20: The radar system of SMAP failed few months after launched. You should specify this, is you talk about the radar system.
This information will be added to the manuscript.

P2.L12: 'Within a snapshot just one or two of the Stokes parameters are measured at the same time'. Please explain this properly, this sentence has no sense. Explain when one or two are measured?
The explanation will be expanded to make it more clear: "When just one of the Stokes parameters is measured, all three arms of the sensor record the same polarization. In the case of recording a cross-polarized snapshot, one arm of the sensor records a polarization while the remaining two arms the other. Measurements of single (XX or YY) and cross-polarization ((XX, XY) or (YY, XY)) are done alternatively."

P2. L23: 'SMAP started providing data starting in April 2015.' Re-phrase.
Sentence will be rephrased.

P.2.L8. It is not clear to me which is the RFI filtering method used in this paper, the one in Huntemann et al 2014, or 2015?

Neither of the two methods are employed for the retrieval that is using the fitted brightness temperatures. The method from Huntemann et al. (2014) was used just for the 40-50 deg retrieval (both 5.05 and 6.20) in Section 3.1. Huntemann et al. (2015) is referenced to show an alternative to the previous filter. It preserves more data improving the overall coverage.

For this paper the RFI filtering for SMOS is done in two steps:

1. using the flags included in the Level 1C SMOS data product as mentioned in L4.P28-29
2. afterwards by the iterative process which eliminates the data with the highest absolute difference from the fitting curve

For SMAP data we use the quality flags for filtering.
The third paragraph of Section 3.2 will be modified to make it clear that the iterative process is the second step used for RFI filtering.

p3.L25: You should explain that a resolution of 12.5km is a subsampling grid, this is not the real SMOS resolution. In fact, in my opinion, it has not sense to go to 12.5Km, only with SMOS data, you should use the grid of 25km. In case you use other data with thinner resolution combined with SMOS then you could go to 12.5km, if not this resolution is completely false. You should explain this clearly in the paper.

A sentence will be added explaining the fact that the SMOS and SMAP spatial resolution is approximately 40 km not 12.5 km. Also the resolutions of the two satellites will be added in the Section 2.
The two reasons for choosing this grid is consistency with the older SIT product which is provided in this resolution and the fact that SMOS L1C product from which we start is already oversampled to 15 km.

Eq. 2: It is very dangerous to use in the equation of Q the exponential of a number which is raised to d (Q=(a-b).exp(-x/c)^d)+d), it makes the model inestable. But I imagine this comes from Huntemman 2014 method...

Yes, the two functions I and Q that were fitted to the training data come from the Huntemann et al., (2014). The parameter d is determined only once, during training. As presented in that paper, the function for Q was chosen empirically to represent well the polarization difference from the training data as dependence on the sea ice thickness.

P3.L28. Please explain – spillover effects. Or Re-word.

Explanation will be added. - The spillover produces an erroneous increase or decrease in brightness temperature of 1 to 1.5~K in the high contrast areas (land/ocean, sea ice edge). The value of the TB change is taken from the SMOS Calibration Team reference.

P4.L9-10. 'Also…version.' Please re-write the sentence. It's unclear.

Will be changed.

P4.L22: 'Moreover,…' , here you are talking about your method to remove RFI and averaging the TB, not on the SMOS acquisition method. This should be specifed, if not, it seems this is a problem of SMOS, since the previous sentence you talk about the SMOS acquisition method. This sentence starting with 'Moreover,' I understand is the basis to explore the new method of ftting a curve, so please explain this clearly.

In this section we tried to explain why using a fit function for TBs of SMOS should improve the results compared with the old algorithm. We considered that:

1. due to the geometry of the snapshot, different incidence angles distributions are available to one pixel along the trackline during an overpass
2. the 300 K threshold RFI filter that will remove whole snapshots.
   These two issues alone or together can generate biases in the resulting averaged incidence angle distribution for a pixel. This issues are eliminated by using a retrieval at a fixed incidence angle which is achieved by using all the data points available for one pixel and fitting them with a curve. See also our answer to your comment regarding P1L5 above. The paragraph will be update with a more clear explanation based on this and the above P1L5 answer.

P5.Eq 3: Are the values of a, b, c, d the ones in table 1? This is not cited here. And are bh and bv from the equation different? I don't see the values of H and V pol in table 1. Moreover, bh and bv, they should be equal, to permit to have Tbh=TBV at Theta=0 (nadir).

Equation 3 represents the brightness temperature fit functions while Table 1 represents the parameters for the Sea Ice Thickness retrieval used in Equation 2. Regarding the two parameters at nadir sin^2 will be 0, eliminating bh and bv parameters while the cosines will be 1 therefor the only remaining value in both equations is C/2 representing the intensity at nadir.

P5.L23. Figure 4 is mentioned in the text before Figure 3. Please correct it.
The change will be done.

P6. L1: You talk about 'estimated retrieval error of 30% of SIT'. This is not commented before, from where do you have this value? Please add reference. Which method has this retrieval error? The daily of the 45º? Please discus better the error budgets as well.

Reference for the 30% retrieval error will be added. The error estimate was computed for Huntemann et al. (2014), it is done for the 40-50 deg daily mean retrieval. Since we consider that the retrieval itself wasn't changed, just the parameters of the retrieval curve (due to the new fitted TBs at a fixed incidence angle that went into the retrieval) the error budget should still be consistent with the retrieval. In addition a new Section discussing the uncertainties in the retrieval, including SIC will be added.

Figures:

Figure 1 and 4: Increase font size.
Font size will be increased.

Figure 5: Please also show the SIT map derived using SMAP only, and SMOS only for the same date alongside SMOS+SMAP SIT map. This will make it easier for visual comparison.

The differences between the SMOS and the SMAP maps are very small and noisy and will not add much information. We will add a difference map between the mixed and the 40 deg SMOS retrieval which will indirectly show the distinctions between the two sensors without cluttering the figure.

References:

-Please add webpage from where to fnd the documents used in the references (Indra Systems and SMOS Calibration…). I am sure they are public.

Links will be added.

SMOS Calibration Team:

https://earth.esa.int/documents/10174/1854503/SMOS_L1OPv620_release_note

The link to the Indra Systems is down for the version cited, an older version which still includes the referenced part is found here (pag. 305):

https://earth.esa.int/documents/10174/1854583/SMOS_L1_Aux_Data_Product_Specification

If the original link will not be restored, the reference will be changed to this one.

-I think the references should order it for name of frist author and year. So reference 3 and 4 should be switch.

The ordering is done automatically by the bibtex style file. My understanding is that multi author papers come after dual author ones regardless of the year of publications thus even if the third reference is from a latter date it will still come before the fourth one. Anyway, we assume that the bibtex style file will produce a correct order as needed by the journal.

Answers to Anonymous Referee  2:

The paper by Paţilea et al. deals with retrieving thin sea ice thickness (SIT) from L-band radiometry using brightness temperatures (TB) from two different satellite missions: SMOS and SMAP. The study mainly deals with three things: 1) An existing SIT retrieval is applied to a newer data version of SMOS. 2) A slightly different approach to fit TBs in the SMOS retrieval is applied (using a TB fit to a specific incidence angle, e.g. 45°, instead of using the average TB from measurements between 40 and 50°). 3) SMAP TBs are "converted" to be used in the SIT retrieval, which was originally set up for SMOS, and combined SMOS and SMAP SIT maps are produced. The produced maps are compared with each other but not compared with independent SIT data.

Unfortunately, the used methods are not described precisely enough (see, for example, comments 6), 10), 11), 12) below). Furthermore, the Level 1B SMAP data is claimed to be "top of the atmosphere" TBs, i.e. atmospheric effects are not taken into account. However, according to the Algorithm Theoretical Basis Document (ATBD): "SMAP Calibrated, Time-Ordered Brightness Temperatures L1B_TB Data Product" [1] and De Lannoy et al. (2015) [2], for example, SMAP Level 1B TB data are "corrected for atmospheric effects". As far as I know, this is also one of the reasons why SMOS (Level 1C) TBs (which are not corrected for atmospheric effects) and SMAP (Level 1B) TBs differ. Thus, it is not clear how the authors dealt with this. In general, I think the differences between SMOS and SMAP should be discussed and presented in more detail.

Regarding TOA TBs of SMOS and SMAP: Yes, SMOS L1C data that is used here do not contain atmospheric correction. While SMAP L1B data files contain both both surface TBs and TBs before correction (including the TB correction it self). This allowed us to choose TOA TBs for SMAP since they are contained in the data files.

Link for data fields in the SMAP  L1B: https://nsidc.org/data/smap/spl1btb/data-fields#toa_v

Section 4.1 dealing with the SMOS/SMAP intercalibration will be overhauled and should explain more clear the differences between the sensors (the possible SMOS bias mentioned in SMOS Calibration Team reference and TOA SMAP TBs are corrected for extraterrestrial noise) in the data from to sensors.

Another issue is that there is a study by Huntemann et al. (2016) [3] that deals with using SMOS and SMAP for ice thickness retrieval and shows a resulting map (it is mainly with the same authors). The paper ([3]) gives the difference between SMAP and SMOS retrieved SIT for Oct to Dec 2015 and contains a SMAP SIT map and the SIT difference between SMAP and SMOS for 7 Oct 2015, while the paper presented here compares the combined SMOS and SMAP SIT with SMOS SIT and shows a combined SMOS and SMAP SIT map and how it differs from SMOS SIT for 11 Oct 2015 (Fig. 5). As these two studies seem to deal with very similar things, I think it is very important to clearly state the difference and the "added value" of the study presented here. My main concern is that there is already a paper ([3]) that shows

how SIT retrieved from SMAP compares with SIT retrieved from SMOS (for the TB polarization difference and intensity approach) for Oct to Dec 2015. The new thing in the study presented here seems to be that SMOS and SMAP SIT are not compared but instead averaged to form combined SIT maps, and these have been compared to SMOS-only maps for Oct to Dec 2015 (I think this is the period, see comment 19)). However, as it is not clear how the SMAP SIT or the combined SMOS and SMAP SIT compare with independent SIT data, just combining the two data sets, as done here, does not necessarily bring new information.

The Huntemann et al. (2016) used the L1C SMAP data, which means gridded and corrected data. In the current paper we used L1B SMAP data which is not gridded and also gives the possibility of choosing Top of the Atmosphere (TOA) TBs which should be closer to the SMOS L1C data which is TOA.

The most important new thing in this paper is the introduction of the fitting function for the SMOS brightness temperatures. This is used for the following things:
1. the iterative process used to determine the fit function parameters in which we removed the highest 20% of data with absolute difference from the fit is replacing the 300 K threshold RFI filter used before. This is additionally to the RFI flags that were included in the new SMOS 6.20 data version and now are used for a preliminary filtering
2. the fit function allows for computing SMOS TBs at a fixed incidence angle. As it is mentioned in Section 3.2 and 3.3 this removes possible incidence angle biases that would have appeared in the 40-50 deg incidence angle averaging, making the retrieval more self consistent and allows computation of a new retrieval curve based on the fixed incidence angle data.
3. regarding SMAP/SMOS calibration and retrieval: in Huntemann et al. (2016) the retrieval curve is it was done using two linear regressions between, first the SMAP TBs were compared 40-50 deg to 38-42 SMOS TBs, to bring SMOS data to an approximate incidence angle of 40 deg.

Section 4.1 will be changed to try to make it more clear what are the differences relative to the Huntemann et al. (2016) paper and improvements that the current paper. Main differences are: here we use TOA TBs, the SMAP data is used for a retrieval trained for 40 deg L-band data (SMAP incidence angle), while in Huntemann et al. (2016) the retrieval was done using the 40-50 deg trained retrieval. The adjustment of SMAP TBs had to take in account also this difference while now, this is not necessary.

The manuscript contains several spelling errors that could have been avoided by simply using a spell checker (thus I will not point them out). Furthermore, the style of writing could be improved and the usage of commas should be revised. Not all statements are supported by references (or a reference is given that I cannot find/access), see specific comments below.

**Specific comments**:
1) Introduction:
- I think the introduction should be improved in that the work presented here is put more into a

scientific context. Is this the first time SMOS and SMAP are merged? What are the "challenges"? Why not elaborate more on the differences between SMOS and SMAP? This is not mentioned at all in the introduction (only shortly in the abstract but that should be independent). In general, I think that the introduction (especially starting from p. 1, l. 22) may be hard to understand for those who are not very familiar with the SMOS SIT retrieval and the SMAP mission.

The differences between SMOS and SMAP will be made clear: fixed vs variable incidence angle range (resulting in different swath size), synthetic vs real aperture, SMAP containing correction for extraterrestrial noize and onboard RFI corrections mechanisms. This issues will be addressed. References to previous attempts for SMAP SMOS TB comparison will be added in introduction.

- No references given for statements made in the first lines of the introduction (p. 1, l. 10-13) and for "The atmosphere has little influence on the radiation at L-band as both absorption and scattering are small." (p. 1, l. 16-17).
Proper references will be added:
"Sea ice as an important climate parameter"
- Moritz et al., (2002), Dynamics of recent climate change in the Arctic, Science, 297, 1497–1502
- Holland et al., (2010), The sea ice mass budget of the Arctic and its future change as simulated by coupled climate models, Clim. Dyn., 34, 185–200,
- Stroeve et al., (2007), Arctic sea ice decline: Faster than forecast, Geophys. Res. Lett., 34, L09501,
"Sea ice inhibits evaporation, reduces heat and gas exchange between ocean and atmosphere and increases the albedo"
- Maykut, G. A.: Energy exchange over young sea-ice in the central Arctic, J. Geophys. Res., 83, 3646–3658, 1978
- Perovich et al., (2012), Albedo evolution of seasonal Arctic sea ice, Geophys. Res. Lett., 39, L08501
"The atmosphere has little influence"
- N. Skou and Dorthe Hoffman-Bang, "L-band radiometers measuring salinity from space: atmospheric propagation effects," in *IEEE Transactions on Geoscience and Remote Sensing*, vol. 43, no. 10, pp. 2210-2217, Oct. 2005
2) Several sentences start with "It" or "This", which can make reading hard because it is not always clear what "it" or "this" refers to (especially if referring to something mentioned in the last sentence or even earlier...).
Example 1 (four sentences in a row): "and it will be used... This is combined ... It is used for... This is also..." (p. 1, l. 24 - p. 2, l. 1)
Example 2: "It adds better ... At the same time it ..." (p. 2, l. 17-18)

The sentences where "it" and "this" add confusion will be changed to clear the confusion.

3) p. 2, l. 9-15, including eq. (1): Maybe this section is not needed because this is not the focus of the study and seems to be the same approach as in Huntemann et al., 2014?

We consider that since for this article all the SMOS data processing on our side starts from the antenna frame of reference. Also for the first step in removal of contaminated RFI observations (Section 3.2 - will add extra sentence for clarity) that is done using the flags contained in the data files: the flagging and removal is done before the antenna to earth reference frame change.

4) Statements on p. 2, l. 17 ("It adds better RFI flagging...") and l. 20-21 ("The new data version also reduces... ") are not supported by references. Is "SMOS Calibration team and Expert Support Laboratory Level 1, 2015" the reference for these? I cannot find or access this. Please provide a web link or other hints how to find this. Also the "Indra Sistemas S.A.: SMOS Level 1 and Auxiliary Data Products Specifications, Product Document, Madrid, 2015" reference may contain more information to make it more easy to find. References for statements in p. 2, l. 29 - p. 3, l. 3?

Will add relevant links for the SMOS Calibration Team and Indra Sistemas documentation. The better RFI flagging is mentioned in the SMOS Calibration Team document at P2-Section 2 and P7-Paragraph g). For the differences between ascending and descending overpasses see Figure 2 from the same document.

SMOS Calibration Team:
https://earth.esa.int/documents/10174/1854503/SMOS_L1OPv620_release_note

The link to the Indra Systems is down for the version cited, an older version which still includes the referenced part is found here (pag. 305):
https://earth.esa.int/documents/10174/1854583/SMOS_L1_Aux_Data_Product_Specification

5) p. 2, l. 18-19: "... it introduces a warm bias in the brightness temperatures of aproximately 1.4 K relative to the previous version 5.05 over ocean..." -> Is 1.4 K warmer brightness temperatures over the ocean more/less realistic? Or just a trade off to get the SMOS brightness temperatures over land (and Antarctica) closer to modeled and measured values? But for this study this is not important anyways because no changes were observed for the high latitudes? This is somewhat confusing…

The information is taken from the SMOS Calibration Team document P4-Section 4-paragraph a). It is suggested that the 6.20 ocean TBs might be approximately 1 K too warm respective to modeled values and 1.4 K warmer relative to the old version. The difference is also seen in Figure 1 (now Figure 2) from the our article, with the intensity for the 6.20 data being higher compared to the 5.05 version, same as in the manuscript. The sentence mentioning the high latitudes is referring just to the ascending/descending differences as they can be seen in Figure 2 of the SMOS Calibration Team document. We will rephrase this paragraph to make it less confusing.

6) p. 2, l. 29 - p. 3, l. 5 two RFI filters are described. However, it is not mentioned which RFI filter approach is used in the study presented here. Later (p. 4, l. 28-30) you describe a third RFI filter method. Were different RFI filters used for different parts of this study?

A third method is used for removing RFI. Section 3.2 will be updated to make it more clear that RFI removal is done in two steps: first by eliminating L1C data that is already flagged with RFI as was already mentioned in (P5L28-30) and second, the data removal in the iterative process is used for eliminating possible remaining RFI influences. (answered more detailed for Referee 1 P2L8 comment)

7) p. 3, l. 12-14: "retrieval based on a fitting function ... through the full incidence angle range. This will provide more stable brightness temperatures and is necessary for a consistently combined SMOS and SMAP ice thickness retrieval." -> What is the "full incidence angle range"? Why is this "more stable"? More stable than method by Huntemann et al., 2014? Is that a result of the work presented here? Or has that been shown somewhere else (where?)? Please clarify. Why is fitting through the full incidence angle range "necessary for a consistently combined SMOS and SMAP" retrieval if SMAP measures only at one incidence angle?

By full incidence angle range we mean that for each grid cell we use the observations from all available incidence angles instead of restricting them to the 40 to 50 deg only as was done in Huntemann et al., 2014.

By using the fit function now we can compute the TB for a grid cell at an exact incidence angle removing the variation of the incidence angle that could appear in the daily averaged observations. Because of the dependence of brightness temperature on incidence angle, a difference in TB between two grid cells can come just due to a difference in the resulted averaged incidence angle. We consider the TBs resulted from fitting more stable due to the removal of the incidence angle effects. This is problem is discussed in the first paragraph of Section 3.2 and 3.3. We removed the sentence in P3L13 because it was adding confusion to Section 3 which doesn't deal with SMAP.

The fitting process allows obtaining SMOS TBs at a fixed incidence angle. This means that we can compute SMOS TBs at 40 deg, the incidence angle of SMAP, removing possible incidence angle biases either for the retrieval itself or during the training of the retrieval curve. Having both TB sets at the same incidence angle makes the combination of them consistent.

8) p. 4, l. 6: "From a total of 5.1 million data points,..." -> Where does this number come from? All Arctic data points from a certain time period?

It represents the cumulated data points from all the 87 days (1 October - 26 December 2010) where we select just grid cells that contain retrieved ice between 1 and 50 cm for at least one of the two algorithms. Sentence will be updated to make it more clear.

9) p. 4, l. 19-22: Maybe a reference to a paper that describes the SMOS snapshot geometry would be useful here.
Reference added - Font et al., - SMOS: The Challenging Sea Surface Salinity Measurement From Space, Proceedings of the IEEE, 98, 649–665, 2010.

10) Where do the equations in eq. (3) come from? How is C "determined by averaging the sum of the polarizations for each observation" (p. 4, l. 33-34)? After all fit parameters

ah,av,bh,bv,dv have been determined first (are they determined for each observation? for each incidence angle range? as generally valid values?) ?

Equation 3 comes from the Zhao paper. We will changed the sentence where the paper is mentioned to refer to the equations (Section 3.2, paragraph 2).

Regarding C: 1. we will correct the manuscript. The correct procedure is not averaging but taking the median 2. for each grid cell we have a large number of observations per day. For each observation we have the TB at two polarization. These two values are summed up for each observations and then we take the median of the result. This represents C. An extra sentence will be added to explain why the median is used and not the mean. As was mentioned in the paper at P5L1 we consider C/2 to be the intensity at nadir.

At each step first C is computed and afterwards the other five parameters are determined through a least squares procedure. This is done separately for the two polarizations with ah, bh for horizontal and av,bv and dv for vertical. One set of six parameters is valid for one individual grid cell for one day. As was mentioned in the paper at P5L8, we don't consider the fit to be optimal for extrapolation that the incidence angle range valid is just in between the minimum and maximum incidence angle contained by the observations.

The paragraph that started at P5L31 was modified to better explain the fitting process.

11) p. 5, l. 9-10: "only grid cells with the incidence angle range of observations that covers the wanted angle, e.g. 45∘, are used for the retrieval" -> What is the exact criterion here? (e.g. "only grid cells that contain at least N observations with incidence angles θ±Δθ around the considered incidence angle θ")

One grid cell contains data points at various incidence angles. We will consider for retrieval just the grid cells that have observations with incidence angles both below and above the required angle for retrieval. If the desired angle is either higher than the observation with the maximum incidence angle or lower than the one with the minimum angle, the grid cell will not be considered for retrieval. There is no minimum incidence angle range around the required angle nor minimum number of observations. The sentence will be modified to make it more clear.

12) p. 5, l. 27-30:
-"On the other hand for many ocean areas which formerly were excluded by the RFI filtering (grey in Fig. 4 left) now data is available, e.g. around Iceland, Eastern Greenland and Vladivostok." -> I don't know which TB measurements are used for the "fitted 45° brightness temperature" method… The text says that you need "at least one observation under 40°" and that you use "only grid cells with the incidence angle range of observations that covers ... 45°" (see also comment above). If the latter means, for example, that you use incidence angles between 40 and 50° PLUS at least one measurement with incidence angle <40° (which adds at least one more possibility to encounter an RFI-contaminated observation), how can this method be less influenced by RFI than the algorithm that uses the daily mean from measurements between 40 and 50°? Is it because of different RFI filters? (see also comment

6)) Is there a minimum number of measurements that has to be available in order to perform the SIT retrieval?
-"At the same time in the area of the Hudson Bay there is a 30% decrease in the area covered due to the high uncertainty of the fit." -> How does the "high uncertainty of the fit" influence the number of grid cells the retrieval is performed on? Is there a criterion for cases in which the retrieval is terminated / not performed? Is that given anywhere in the manuscript?
- Is the same number of SMOS measurements used for both retrieval approaches here? SMOS data can be quite "scattered", the number of data points used/averaged in the retrieval can have an impact on the results…

Most of Section 3.3 will be rewritten making it more clear what is calculated.

- "I don't know which TB measurements are used for the "fitted 45° brightness temperature" method…". For each grid cell we use all the data points at all incidence angle available and we used this data to compute two fit curves which represent the dependence of the TB on the incidence angle (this is done for each polarization separately). After we obtain the parameters of the fits we used them to compute the TB at the 45 deg incidence angle. This brightness temperature is then used to compute the SIT.

- "The text says that you need "at least one observation under 40°" and that you use "only grid cells with the incidence angle range of observations that covers ... 45°" (see also comment above)". 1. Since we assume that the two polarizations vary synchronous with the incidence angle (TBv increasing and TBh decreasing from nadir) until approximately 40 deg ; this is considered together with 2. the C parameter in the fit function is computed using just under 40 deg incidence angle data (Tian-Kunze et al. (2014) - Fig. 3 TB vs TBh/2+TBv/2). If the we would have just higher incidence angle data, even if the data might not contain any RFI influences the asynchronous variation of the two polarizations would bias C/2 which we consider to be the intensity at nadir.

- In our current case the SIT will be computed for one grid cell just if we have at least one data point below 40 deg and at least one data point at 45 deg or higher (due to the necessity not to extrapolate the fit)

- "The text says that you need "at least one observation under 40°" and that you use "only grid cells with the incidence angle range of observations that covers ... 45°" (see also comment above)" - we consider that RFI sources although can propagate through a snapshot don't need to contaminate all of it. The simple threshold of 300 K used as a filter was removing possibly a lot of unaffected data. In the Huntemann et al. (2015) a threshold for the standard deviation for incidence angle binned data eliminated RFI while preserving more data. We consider that by removing data points with high difference from the fit function in the iterative process we can remove RFI without removing too much of the unaffected data.

- "How does the "high uncertainty of the fit" influence the number of grid cells the retrieval is performed on?" Sentence corrected. The grid cells in Hudson Bay don't fulfill at least one of the two thresholds mentioned (at least one data point below 40

deg and and one above or equal to 45) and also the least square routine can fail to converge on a solution for the fit parameters hence we cannot compute the TBs at the desired incidence angle. The 40 deg data requirement also means the overall the swath will be narrower resulting in larger areas not covered at low latitudes.

- "Is the same number of SMOS measurements used for both retrieval approaches here? SMOS data can be quite "scattered", the number of data points used/averaged in the retrieval can have an impact on the results…" - that is one of the benefits of using the full incidence angle range. A fit procedure will use all the data available from all incidence angles and at the and using the resulted parameters of the fit to compute the TB at a fixed angle that is used for the retrieval. Meanwhile the daily average due to the strict incidence angle range for TBs can end up having a biased average incidence angle and as a result TBs and SIT. For a fit we can have many data points below 40 and a few close to 50. We expect that in a case like this the fit will return TBs closer to the real value at 45 deg than the daily mean approach that will end up with an incidence angle average close to the high end of the range.

13) In the manuscript text, Fig. 4 is mentioned before Fig. 3.

Will be changed.

14) p. 6, l. 1: "estimated retrieval error of 30% of SIT." -> Where does this number come from?

Huntemann et al. (2014), Reference will be added.

15) p. 6, l. 5: "within the error margin of the retrieval" -> Which error margin? From where? (the 30% given in l. 1?)

Yes, the 30% from Huntemann et al. (2014). Sentence will be changed.

16) Do the RMSD and bias given on p. 6, l. 2-5 refer to the comparison of the two retrieval approaches for just the one day (29 Oct 2010) shown in Fig. 3 and 4? I wouldn't consider this representative…

Yes, this was done. We will keep Fig. 4 as an example of one day, and will calculate the statistics for the ~3 months of the 2010 freeze-up. A double figure will be added with SIT bias and RMSD for each individual day and, for the whole period for 1 cm thickness bins.

17) p. 6, l. 16-17: You mention that Lannoy et al. (2015) take into account atmospheric contributions for SMOS "to convert between SMOS and SMAP TBs". You didn't, did you? (earlier (p. 2, l. 26) you claimed that SMAP TBs are "top of atmosphere"). Please write clearly what YOU have done to bring SMOS and SMAP TBs together and how and why your approach differs from other approaches etc.

No, we have not corrected for atmospheric contributions. We only mention Lannoy et al. (2015) as another attempt to convert between SMOS and SMAP TBs before ours.

SMAP L1B datafiles contain both surface and TOA brightness temperatures.
The Section 4.1 where we explain the calibration has been extended to make it more clear how we actually bring the SMAP TBs to SMOS TBs and how is it different from the previous approach (see also answer to last General comment).

18) p. 7, l. 1-2: "For 11 October 2015 (not shown) the differences ... between SMOS fitted TBs retrieval and SMAP retrieval are small." Why do you pick one day to state this, and then not even show it? What about the other days? What is "SMOS fitted TBs retrieval" here? The one fitted to 40° incidence angle?

The SIT is calculated using TBs from SMOS at 40 deg. For SMOS vs SMAP, statistics will be calculated for the whole 3 months period. The Section will be updated with the appropriate statistics. Figure 5 will include also a SMOS@40 deg - SMOS+SMAP subfigure as an example from which we can infer the contributions of SMOS and SMAP to the merged product.

19) p. 7, l. 13-14: "The RMSD between the original 40∘ to 50∘ incidence angle daily mean retrieval from Sect. 3.1 and the new mixed sensor one is 2.14 cm for the three months period investigated, while the bias is -0.63 cm..."
-What is the "three months period investigated" here? The same as was used for calibration earlier (Oct to Dec 2015)?
-It would be interesting to know the difference between SIT from SMOS fitted to 40° and SMOS+SMAP; otherwise we cannot distinguish whether the difference comes mainly from the different approaches for the SMOS retrieval (fitted to 40° or mean between 40 and 50°) or the different sensors (SMOS or SMOS+SMAP). Indeed, the values from the comparison in 3.3 are very similar (bias 0.69cm and RMSD 2.2cm), although a) I am not sure in which direction the negative sign in "-0.63 cm" points here, and b) the results in 3.3 seemed to be for a comparison of one day only.

Yes, the three months period is the same period that was used for calibration. The bias and RMSD for SMOS vs Merged dataset will be included in this section to differentiate better between the contributions of the two sensors. b) the comparison in 3.3 will be extended to a three months period to make it more relevant.

20) In the conclusions, there seems to be a "new" result: "Using SMAP data and the SMOS data fitted to the same incidence angle to calibrate the SMAP TBs to those of SMOS has improved the TB RMSD between the two datasets for both polarizations with 2.7 K and 2.81 K for TBh and TBv, respectively." (p.7, l. 22-24) -> This was not mentioned before. Do you mean improved to or by 2.7K? The sentence is hard to understand, also because the reference to the "improvement" comes only in the next sentence. It says "This is an improvement from previous attempts (Huntemann et al., 2016) where the the RMSD for both polarizations was over 4 K showing that using fixed incidence angles for SMOS data increases the accuracy." The phrasing could be clearer (who concludes this?). And I don't agree... 1) This shows only that SMOS and SMAP TBs as compared here and in Huntemann et al. 2016 agree better if a fixed incidence angle is used. And as I don't know how you dealt with the atmospheric corrections for SMOS, I cannot say whether it is a good thing that the

SMOS TBs you used here (not corrected for atmosphere? corrected differently than SMAP?) and the SMAP TBs (atmosphere corrected) are closer to each other or not... 2) Averaging SMOS TBs differently does not "increase the accuracy" of SMOS data, but these SMOS data might be more suitable for comparison with SMAP.

The two values (2.7 and 2.81 K) are the RMSD of the linear regression between SMOS at 40 deg and SMAP TBs. This values are improved relative to the Huntemann et al. (2016) paper where they were around 4 K. We will add the relevant sentences to the calibration section and a new Figure with the density plots and regression line.

"This shows only that SMOS and SMAP TBs as compared here and in Huntemann et al. 2016 agree better if a fixed incidence angle is used." - Since during the calibration the SMAP TBs are adjusted to the SMOS ones, we consider that a better agreement between the two datasets will reduce the uncertainty introduced by the adjustment.

As mentioned in previous comments the SMAP L1B data we use contains also TOA brightness temperatures which were used here.

"2) Averaging SMOS TBs differently does not "increase the accuracy" of SMOS data, but these SMOS data might be more suitable for comparison with SMAP."

21) What about ice concentration? Has big impact on L-band TBs but is not mentioned at all
.
Section 5 will be added for discussing uncertainties estimates, including SIC. The change of SIT with SIC will be discussed, and some estimates of the error introduced by SIC will be provided. Also a sentence is added in Section 3.1 regarding the usage of SIC in the training of the retrieval (P3L19). The training using SIC follows the Huntemann et al. (2014) algorithm.

22) The numbers given in Tab. 2 for the "original 5.05 data version" (SMOS) do not correspond to the numbers given in the literature for this SIT retrieval with SMOS (Huntemann et al., 2014 & Huntemann et al., 2016). (a and b interchanged for I_abc; different a, c, d for Q_abc).
Thank you for catching this discrepancy. The values have been corrected for I. The parameters used in the manuscript are the ones  currently used for the operational SMOS ice thickness retrieval at University of Bremen for SMOS data version 5.05. They are different from the ones in Huntemann et al. (2014) because a different SMOS data version was used there. This information was added to the text under Section 2 P2L22.

23) Fig. 1 in the current print version is quite small, especially the label font size.
Will be changed.

**Further comments**:

p. 1, l. 4-5: "... SMAP observes at a fixed 40∘ incidence angle which makes thin sea ice thickness retrieval more stable as incidence angle effects do not have to be taken into account." -> Why would this be "more stable"? Isn't it mainly a restriction to not have any measurements at incidence angles other than 40°? For example, wouldn't a retrieval using only SMOS measurements at 40° be as stable?

We have changed "stable" to "consistent".
Yes, as we used "stable" in this context, we meant exactly that: a fixed incidence angle retrieval is stable compared to averaged observations where the mean incidence angle might vary. Thus a 40 deg SMOS retrieval is as stable as a SMAP one in this context.

p. 1, l. 20: Very abrupt and sudden transition from describing SMOS to introducing SMAP.
The SMAP introduction was rephrased and moved to the next paragraph.

p. 1, l. 22-23: "... by calibrating the SMAP brightness temperatures (TBs) to those of SMOS" -> Why do SMAP measurements have to be calibrated to SMOS measurements? Here you could mention whether/how they differ. I guess this has also to do with SMOS brightness temperatures having been compared (or "validated") with other sources (in Huntemann et al., 2014, for example).

The algorithm was developed for SMOS data and the results were compared with other sources. Since the algorithm is empirical, we consider that the retrieval curve obtained is valid in a direct manner just for SMOS observations. In Section 3.1 we presented that even for SMOS data only we can have biases (currently Fig. 2) that appear with a new data version, we consider that we can not apply the algorithm directly to SMAP data.

Although measurements for both satellites are provided at the top of the atmosphere, SMAP L1B TOA data contains galactic noise and sun specular reflection corrections while SMOS does not (was mentioned in the calibration section P6L28-29). Also, a possible 1 K warm bias of the ocean is mention in the 6.20 data release notes (value will added to P2L18).

Sentence will be added in introduction: "Calibration is necessary due to a possible warm bias in SMOS data (Sect.~\ref{sources}) and due to corrections for galactic noise and sun specular reflection in SMAP data.". Furthermore, Section 3.3 dealing with the intercalibration will be expanded to explain the differences between SMOS and SMAP TBs.

p. 2, l. 12: "the the"
p. 2, l. 16: "... version 6.20 has been available since 5 May 2015 operationally..." -> version 6.20 has been operationally available since 5 May 2015
p. 2, l. 24: "a equator" -> an equator
p. 2, l. 24: "...data is used, which contain..." -> "...data is used, which containS..." or "...data ARE used, which contain..." (but then "data" has to be used as a plural word throughout the paper)
p. 2, l. 27: "36x47 km" -> 36 km x 47 km
p. 3, l. 20: "retrieval curve" singular vs. two equations
All comments were implemented.

p. 4, l. 7: "it increases" -> they increase?

Rephrased to "The bias and standard deviation are under ±1 cm and 2 cm, respectively, for ice thicknesses below 25 cm. For 50 cm thickness the bias increases to +4 cm while the standard deviation reaches 11~cm.

p. 4, l.10: "This allows to estimate" -> I think this is grammatically incorrect (while "This allows us to estimate..." should be ok I guess)
p. 4, l. 13-14: Remove "given above"?
p. 4, l. 29: "observation" -> observations
p. 7, l. 17-18: I recommend to combine these two lines to one because they are very repetitive.
p. 7, l. 19: "rage" -> range
p. 7, l. 25: "the the"
The modifications will be included in the manuscript.

References:
[1] Algorithm Theoretical Basis Document (ATBD): "SMAP Calibrated, Time-Ordered Brightness Temperatures L1B_TB Data Product", J. Piepmeier, P. Mohammed, G. De Amici, E. Kim, J. Peng, and C. Ruf (2014), e.g. Tab. 1, p. 11, file accessed at
https://www.google.de/url?
sa=t&rct=j&q=&esrc=s&source=web&cd=3&ved=0ahUKEwiF2sqb3MDYAhVBa1AKHbJKAflQ
Fgg2MAI&url=https%3A%2F%2Fsmap.jpl.nasa.gov%2Fsystem%2Finternal_resources%2Fde
tails
%2Foriginal%2F278_L1B_TB_RevA_web.pdf&usg=AOvVaw34MP0zjMXDqHm83OB1qAog

[2] De Lannoy, G. J., Reichle, R. H., Peng, J., Kerr, Y., Castro, R., Kim, E. J., & Liu, Q. (2015).
Converting between SMOS and SMAP level-1 brightness temperature observations over nonfrozen land. IEEE Geoscience and Remote Sensing Letters, 12(9), 1908-1912.
[3] Huntemann, M., Patilea, C., and Heygster, G. (2016): Thickness of thin sea ice retrieved from SMOS and SMAP, in: Proceedings of 2016 IEEE International Geoscience and Remote Sensing Symposium (IGARSS), pp. 5248–5251

---

## Referee Report (RR1)

The revised manuscript by Patilea et al. has improved regarding clarity and writing and also the differences to the paper by Huntemann et al. (2016), which covers a very similar topic, are now made clearer. However, I still have major concerns regarding the paper:

**1)** Compared to the study by Huntemann et al. (2016), the authors here
a) use a newer version of SMOS data, which leads to a mean deviation in sea ice thickness (SIT) of 0.22cm & RMSD=1.35cm for TBs in the Arctic for 1 Oct - 31 Dec 2015 (this time period is also used for the following comparisons).
b) use a slightly different RFI filter and choose the SMOS TBs somewhat differently than before: They fit the SMOS TBs to 45deg incidence angle instead of using the 40-50deg TB mean. This results in a mean deviation of -0.3cm & RMSD=2.0cm compared to a).
c) fit SMOS TBs to 40 deg incidence angle (which is the incidence angle provided by SMAP) and perform a linear regression between SMOS and SMAP TBs to apply the SIT retrieval to SMAP , which does appear to be a more "straightforward" approach for combining SMOS and SMAP for the SIT retrieval as compared to the approach in Huntemann et al. (2016). The SMOS and SMAP SITs differ on average by 0.2cm with RMSD=2.4cm. The mean deviation between the combined SIT product and the SMOS SIT derived from 40deg incidence angle is <0.1cm & RMSD=1.4cm.

>Conclusion: The modifications made to the already existing retrieval seem to be relatively small, and as none of these data sets is compared to independent data, the presented modifications to the joint retrieval from SMOS and SMAP may be more of "technical importance" showing small changes (theoretical improvements?) to a previous suggestion to combine SMOS and SMAP data. From the presented study it is also not clear whether there is any advantage in using a combined SMOS and SMAP SIT retrieval as compared to the existing SMOS-based retrieval. By using TBs at 40deg incidence angle instead of 45deg, the usable polarisation difference range for the retrieval reduces from Delta_TB=32K (22...54K) to Delta_TB=26K (17...43K), which can be a disadvantage. On the other hand, as one of the advantages the authors claim that the data coverage by the joint product is 6% larger (for the area north of 55.7deg N). However, from Fig. 6, I would infer that the gain in data coverage is mainly achieved at the edge of the selected area, i.e. in areas that are covered by ocean and not sea ice, which would make them somewhat irrelevant for the SIT maps...

**2)** In the revised manuscript version, the authors have added a section and new figures on the uncertainty assessment. For the uncertainty computation, the authors assume that TB uncertainty of SMOS is equal to the RMSD resulting from fitting the SMOS TBs to 40deg incidence angle. Firstly, as the authors also recognize, this is an underestimation because the RMSD in the fitting iteration is limited to 5K. And secondly and more importantly, this is NOT the only source of TB uncertainty! The relationship between SIT and TB depends on the ice conditions: For example, the salinity of the ice (or more precisely: brine volume fraction), the snow cover, the ice type (these are not mentioned at all), and the ice concentration (mentioned later by the authors but not used to estimate the uncertainty). This variability (and thus uncertainty) is (partly) reflected in the scatter of the training data for the retrieval curve (only partly because the training was done only for Oct-Dec 2010 and in two specific areas of the Arctic). The variability of TBs at the same place and time at different incidence angles should only reflect a small part of the uncertainty...

Fig. 7 shows SIT as function of TBH and TBV, although in the retrieval, SIT is a function of polarisation difference and intensity. This makes the figure and its implications hard to interpret. It is also not clear which of the shown combinations of TBH and TBV (or better: polarisation difference and intensity) are actually seen in the satellite observations. This raises also the question how the retrieval is actually performed using the retrieval curve. In Huntemann et al. (2016), I found: "The minimum euclidean distance of the fit to the data in the I-Q-space defines the retrieved ice thickness." Is the retrieval performed no matter how large this "minimum euclidean distance" is?

If so, how representative are retrieved SIT with large distance from the retrieval curve? What is the distribution of distance values encountered during the retrieval? In Fig. 7, I think, we can see how not restricting the "minimum euclidean distance" to a maximum value leads to an odd behaviour, as seen for TB combinations below the TBH-TBV 1:1 line (where a small change leads to completely different SIT to be retrieved).
Also, from the text (Sect. 3.2) I understand that the retrieval is performed when at least two TB values are found (at least one below 40 deg and one above 45 deg). This issue is, for example, discussed in the paper by Schmitt et al. (2018), which also deals with combining SMOS and SMAP data for sea ice applications. They perform the incidence angle fit only if at least 15 measurements are found, which seems more appropriate considering the relatively high scatter of SMOS data...

> Conclusion: I do not agree with how the uncertainty is estimated and I would suggest to show sensitivity of polarisation difference and intensity instead of TBH and TBV for the values actually encountered during the retrieval, which have to be analysed/shown first.

**3)** Another issue is that the authors have added some more statistics based on a three month period (Oct-Dec 2015) as compared to presenting mainly statistics based on one day of data as was done in the first manuscript version. However, in the revised version, there is still conclusions based on the analysis of one single day of data (Sect. 3.3 & 5.1). I think, example maps for one day can be ok/useful, but, as far as I can see, statistics based on one day of data are not necessarily useful (unless their representativity is shown or at least discussed).
In addition, the authors use the expression "bias" for comparisons of data sets (e.g. SMOS v5.05 vs. v6.20 SIT retrieval in Sect. 3.1, fitted to 45 deg vs. 40-50deg average SMOS SIT in Sect. 3.3 and SMOS vs. SMAP in Sect. 4.1). As far as I know, "bias" is not used (and is indeed very confusing) for comparisons of two data sets, of which it is not clear which one is more realistic/better etc.). I think "mean difference" would be more appropriate here.

**Further comments:**

-p. 1, l. 4: "SMAP observes ... which makes thin sea ice thickness retrieval more consistent" -> This sounds like SIT retrieval from SMAP is more consistent than from SMOS, while you probably aim to say (as we have already discussed and agreed on in the first review round) that the retrieval is "easier" (or more consistent if you like) if a fixed incidence angle is used instead of an incidence angle range, which is, in principle, also possible with SMOS data (by choosing TBs at this incidence angle only). This should be expressed unambiguosly.
- p. 2, l. 28: "Its resolution is..." -> better: The grid spacing is...
- p. 4, l. 2: And which RFI filter are you using? Either write it here or refer to where you describe this in the paper.
- p. 5, l. 11-13: First, you write about "bias" and "RMSD", then about "bias" and "standard deviation". Are RMSD and standard deviation the same here?
- p. 6, l. 3 : Reference to "Eq. 3" is misleading here. Eq. 3 has not been given yet and could mean that you refer to Eq. 3 in Zhao et al. (2015). Also at p. 6, l. 8: First give Eq. 3, then refer to it.
- p. 6, l. 8-18: It sounds like C is determined first and then only the other parameters (ah, bh,...) are determined? Is this / how is this done? Are ah, bh, ... determined differently than C?
- p. 6, l. 19-20: To make this clearer, I suggest to add something like: In this case, the least squares method to fit the parameters is repeated.
- p. 6, l. 8-24: It would be helpful to clearly state that you do not take into account the measurements that do not meet the defined criteria after a maximum of five iteration steps (if this is how these cases are handled...).
- p. 6, l. 28 & p. 7, l. 4: You probably refer to Fig. 1 here instead of Fig. 2.
- p. 7, l. 20: "This is ..." -> What is "this" here? (bias and/or RMSD?)

- p. 7, l. 20- 23: Complicated/unclear description of what you do here... E.g. "selecting all grid cells with that thickness..." -> what is "that" thickness?
-p. 7, l. 22-23: " Only grid cells that contain at most 50 cm and non-zero in at least one of the two algorithms are used." -> Make it clearer whether the condition "in at least one of the two algorithms" applies also to the selection of grid cells with at most 50 cm SIT.
- p. 7, l. 27 & l. 28: "always generating" & "will generate" -> For clarity, maybe add "falsely"/'spuriously".
- p. 7, l.35: What is the "absolute bias" (as compared to just "bias"?)
- p. 9, l. 6: Does the reference to "(Sect. 4.1) refer to Sect. 4.1 in Huntemann et al. (2016) or what is meant here?
- p. 11, l. 1-2: "a weighted standard deviation... is used" -> weighted for what?
- p. 11, l. 3: "The correlation was calculated for a period of seven days." -> Why only seven days?
- p. 11, l. 22-24: Not very clear description.
- p. 11, l. 30-31: "As we have observations at two polarizations at each grid cell available, it should in principle be possible to retrieve SIT and ice concentration simultaneously." -> "In principle", this is only possible if TB varies independently with ice concentration and ice thickness, which would have to be shown before stating phrases like this.
- Sect. 5.2: Why are these considerations not used to estimate SIT uncertainty?
- throughout the paper:
a) Use "an" instead of "a" before abbreviations RFI, RMSD, L-band.
b) The text still contains some typos / language issues (e.g. mixed usage of singular and plural).
c) TB is introduced as abbreviation for brightness temperature, SIT for sea ice thickness, but sometimes the authors use these abbreviations, sometimes not (can even vary within one sentence).
-Is the area used for training and inter-calibration of SMOS and SMAP data defined somewhere in the paper?
-no (clear) references for statements on p. 1, l.15; p. 2, l. 4-8; p. 3, l. 5-8 & l. 13 & l. 14 & 16-19.

**Reference:**
Schmitt, A. U. and Kaleschke, L. (2018): A Consistent Combination of Brightness Temperatures from SMOS and SMAP over Polar Oceans for Sea Ice Applications, Remote Sensing, Vol. 10 (4), 553.

---

## Author Response (AR2)

**Referee #3**
**Suggestions for revision or reasons for rejection (will be published if the paper is accepted for final publication)**Second Review
Authors: Patilea et al.
Title: Combined SMAP/SMOS Thin Sea Ice Thickness Retrieval
M.S. number: tc-2017-168

I thank authors for providing responses to the queries and accordingly editing the manuscript. The revised version is much improved now.

I am glad to see a new section 5 describing uncertainties and SIC impact on SIT. However, on the validation of the proposed combined retrieval, I could not find it in the manuscript. I agree with authors that not much in situ SIT data are available for validation, but there were experiments such as SMOSIce where such data can be available for validation. It is difficult to assess the performance of the retrieval method without validation!
Section 6  (P14) has been introduced. Here we compare ship observations with SIT from the mixed product.

The mere assumption of 100% SIC is insufficient for the analysis. The SIT data must correspond to actual 100% SIC. In Figure 9, the assumed SIT on x-axis should be replaced by the available in situ SIT. Authors can also discuss, based on in situ observations (not the assumed SIT) what % of SIC corresponds to only thin sea ice.
Through Figure 9 and its associated explanations we tried to show that SIC has a big impact on the SIT retrieved, all in a theoretical framework. This was included so that it should be taken in account if the SIT data is used. Also, we provided references to papers (Ivanova et al., 2015, Tian-Kunze et al., 2014) that uncertainties in the SIC retrievals if used for a correction of the current algorithm could induce higher error than just keeping the assumption of 100% sea ice. Although a correction for SIC is not implemented, the uncertainty for SIC in the generating the retrieval curve is considered and this will be added to the SIT uncertainty already presented (paragraph starting P13L22 and Fig. 9).

P1.L26. Retrieval of what? Please use: 'To date, ice thickness retrieval algorithms…'
Done P2L1-3

P5.L8. What do you mean by 'spillover'? What is it? Provide references, if possible. Explain in details or delete.
We added extra explanation and three references to the problem of land-sea contamination (spillover) problem of SMOS. P5L12-16

P6.L28. Retrieval curve in Fig. 2 green, I cannot find it.
P7.L4. Fig. 2 black. I cannot find it.
Done. It was an typing error, it is Fig. 1

Throughout the paper: 'data' is a plural word, use verb accordingly.

Authors must provide a clear rationale for the need of merging SMAP and SMOS. Why is it needed to combine two data sets when one can have SIT products from each separately? The current algorithm allows for both. The data that will be released include SMOS, SMAP and mixed SIT and uncertainties provided separately. This is already mentioned in the conclusions: The algorithm was transferred to the 40 deg incidence angle not just so that both satellites are used together, but also so that the retrieval can be applied just to SMAP.

At the same time SMAP was calibrated to the SMOS TBs so that we can have two sources of TBs for the same algorithm, providing a continuous time series of SIT data starting in 2010 in case SMOS stops functioning after eight years in orbit.

Please be specific with the methodology, how you propose to achieve the objective of the paper in view of the above rationale.
Please write down the objectives of the paper under bullets.
We introduced an algorithm summary in Sect. 4.2.1 that should cover this comment. It describes the complete process of transforming the SMOS and SMAP data into mixed SIT and their uncertainty.

**Referee #2**

The revised manuscript by Patilea et al. has improved regarding clarity and writing and also the differences to the paper by Huntemann et al. (2016), which covers a very similar topic, are now made clearer. However, I still have major concerns regarding the paper:

**1)** Compared to the study by Huntemann et al. (2016), the authors here
a) use a newer version of SMOS data, which leads to a mean deviation in sea ice thickness (SIT) of 0.22cm & RMSD=1.35cm for TBs in the Arctic for 1 Oct - 31 Dec 2015 (this time period is also used for the following comparisons).
b) use a slightly different RFI filter and choose the SMOS TBs somewhat differently than before:
They fit the SMOS TBs to 45deg incidence angle instead of using the 40-50deg TB mean. This results in a mean deviation of -0.3cm & RMSD=2.0cm compared to a).
c) fit SMOS TBs to 40 deg incidence angle (which is the incidence angle provided by SMAP) and perform a linear regression between SMOS and SMAP TBs to apply the SIT retrieval to SMAP , which does appear to be a more "straightforward" approach for combining SMOS and SMAP for the SIT retrieval as compared to the approach in Huntemann et al. (2016). The SMOS and SMAP SITs differ on average by 0.2cm with RMSD=2.4cm. The mean deviation between the combined SIT product and the SMOS SIT derived from 40deg incidence angle is <0.1cm & RMSD=1.4cm.

>Conclusion: The modifications made to the already existing retrieval seem to be relatively small, and as none of these data sets is compared to independent data, the presented

modifications to the joint retrieval from SMOS and SMAP may be more of "technical importance" showing small changes (theoretical improvements?) to a previous suggestion to combine SMOS and SMAP data.

From the presented study it is also not clear whether there is any advantage in using a combined SMOS and SMAP SIT retrieval as compared to the existing SMOS-based retrieval. By using TBs at 40deg incidence angle instead of 45deg, the usable polarisation difference range for the retrieval reduces from Delta_TB=32K (22...54K) to Delta_TB=26K (17...43K), which can be a disadvantage. On the other hand, as one of the advantages the authors claim that the data coverage by the joint product is 6% larger (for the area north of 55.7deg N). However, from Fig. 6, I would infer that the gain in data coverage is mainly achieved at the edge of the selected area, i.e. in areas that are covered by ocean and not sea ice, which would make them somewhat irrelevant for the SIT maps

We aimed towards a more robust retrieval where the coverage is just a minor point. The major improvement of a dual sensor product we attribute to the more representative estimation of a daily average ice thickness, both regarding the radiometric stability of the signal and the variability of the surface conditions regarding day/night cycles (even though sunlight is basically absent in the polar night) as well as spatial variability of sea ice, i.e., drift within the 24h timespan of a day.

"advantage in using a combined SMOS and SMAP SIT retrieval as compared to the existing SMOS-based retrieval" - although the final objective of the manuscript is to obtain a mixed product, the changes to the initial algorithm allow also for: SMAP only retrieval and SMOS 45 deg retrieval which won't be affected by the decreased dynamic range.

Regarding the decreased dynamic range of the polarization in the 40 deg new retrieval: Although the decrease of the dynamic range could increase the sensitivity of the retrieval to small changes in Q, the non-linear change in the curve, with a big decrease at small thicknesses (approx. 11 K at 0 cm) and a smaller decrease in dynamic range at the higher thicknesses (approx. 5 K) means that the highest impact of the diminished dynamic range is at small thicknesses, an area of the retrieval which is more dependent on the change in intensity and not in polarization difference. Changes at P8L23-27.

**2)** In the revised manuscript version, the authors have added a section and new figures on the uncertainty assessment. For the uncertainty computation, the authors assume that TB uncertainty of SMOS is equal to the RMSD resulting from fitting the SMOS TBs to 40deg incidence angle. Firstly, as the authors also recognize, this is an underestimation because the RMSD in the fitting iteration is limited to 5K. And secondly and more importantly, this is NOT the only source of TB uncertainty!

The relationship between SIT and TB depends on the ice conditions: For example, the salinity of the ice (or more precisely: brine volume fraction), the snow cover, the ice type (these are not mentioned at all), and the ice concentration (mentioned later by the authors but not used to estimate the uncertainty). This variability (and thus uncertainty) is (partly) reflected in the scatter of the training data for the retrieval curve (only partly because the training was done only for Oct-Dec 2010 and in two specific areas of the Arctic). The

variability of TBs at the same place and time at different incidence angles should only reflect a small part of the uncertainty…

Fig. 7 shows SIT as function of TBH and TBV, although in the retrieval, SIT is a function of polarisation difference and intensity. This makes the figure and its implications hard to interpret. It is also not clear which of the shown combinations of TBH and TBV (or better: polarisation difference and intensity) are actually seen in the satellite observations. This raises also the question how the retrieval is actually performed using the retrieval curve. In Huntemann et al. (2016), I found: "The minimum euclidean distance of the fit to the data in the I-Q-space defines the retrieved ice thickness." Is the retrieval performed no matter how large this "minimum euclidean distance" is?If so, how representative are retrieved SIT with large distance from the retrieval curve? What is the distribution of distance values encountered during the retrieval? In Fig. 7, I think, we can see how not restricting the "minimum euclidean distance" to a maximum value leads to an odd behaviour, as seen for TB combinations below the TBH-TBV 1:1 line (where a small change leads to completely different SIT to be retrieved).
Also, from the text (Sect. 3.2) I understand that the retrieval is performed when at least two TB values are found (at least one below 40 deg and one above 45 deg). This issue is, for example, discussed in the paper by Schmitt et al. (2018), which also deals with combining SMOS and SMAP data for sea ice applications. They perform the incidence angle fit only if at least 15 measurements are found, which seems more appropriate considering the relatively high scatter of SMOS data…

> Conclusion: I do not agree with how the uncertainty is estimated and I would suggest to show sensitivity of polarisation difference and intensity instead of TBH and TBV for the values actually encountered during the retrieval, which have to be analysed/shown first.

The (TBh, TBv) sensitivity plot (Fig. 7 - currently Fig. 8) has been changed to a (Q,I) one. The errors are now computed relative to the (Q,I) space making a more clear connection between the retrieval curve and the sensitivity of SIT to the change in TBs.

"is the retrieval performed no matter how large this "minimum euclidean distance" is?" - Yes, the retrieval is performed no matter the actual distance from the curve.

"Also, from the text (Sect. 3.2) I understand that the retrieval is performed when at least two TB values are found (at least one below 40 deg and one above 45 deg). This issue is, for example, discussed in the paper by Schmitt et al. (2018), which also deals with combining SMOS and SMAP data for sea ice applications. They perform the incidence angle fit only if at least 15 measurements are found, which seems more appropriate considering the relatively high scatter of SMOS data…" - Due to the restrictions we have for the 40 deg retrieval, at least one value under 40 deg and at least one over 40 deg, and due to the acquisition geometry of SMOS, i.e. the shape of the snapshots, many more observations are taken at different incidence angles. Short statistics for a single day show that more than 97% of the grid cells fulfilling aboves conditions covered have at least 15 data points.
 (Fig. 6 from J. Font *et al*., "SMOS: The Challenging Sea Surface Salinity Measurement From Space," in *Proceedings of the IEEE*, vol. 98, no. 5, pp. 649-665, May 2010.)

**3)** Another issue is that the authors have added some more statistics based on a three month period (Oct-Dec 2015) as compared to presenting mainly statistics based on one day of data as was done in the first manuscript version. However, in the revised version, there is still conclusions based on the analysis of one single day of data (Sect. 3.3 & 5.1). I think, example maps for one day can be ok/useful, but, as far as I can see, statistics based on one day of data are not necessarily useful (unless their representativity is shown or at least discussed).

In addition, the authors use the expression "bias" for comparisons of data sets (e.g. SMOS v5.05 vs. v6.20 SIT retrieval in Sect. 3.1, fitted to 45 deg vs. 40-50deg average SMOS SIT in Sect. 3.3 and SMOS vs. SMAP in Sect. 4.1). As far as I know, "bias" is not used (and is indeed very confusing) for comparisons of two data sets, of which it is not clear which one is more realistic/better etc.). I think "mean difference" would be more appropriate here.

Section 3 contains Figure 3 so that we can show visually and discuss some of the changes that can appear for one day (as also mentioned by the referee). But in Figure 4 (and the discussion of the figure in the paper) the statistics are computed for a 3 months period that covers the freeze up period of 2010 (discussed in the last paragraph of Sect. 3.3). The term "bias" is indeed misleading and it is replaced by "mean difference" as suggested. For Sect. 5.1 (now 5.2) we included the plots to give an example how the uncertainties will look, not for long term statistics. The current day that was chosen, contains a good distribution of intermediate SIT values.

**Further comments:**

-p. 1, l. 4: "SMAP observes ... which makes thin sea ice thickness retrieval more consistent"
-> This sounds like SIT retrieval from SMAP is more consistent than from SMOS, while you probably aim to say (as we have already discussed and agreed on in the first review round) that the retrieval is "easier" (or more consistent if you like) if a fixed incidence angle is used instead of an incidence angle range, which is, in principle, also possible with SMOS data (by choosing TBs at this incidence angle only). This should be expressed unambiguously.

Sentence rephrase to make it less ambiguous. P1L4-6

- p. 2, l. 28: "Its resolution is..." -> better: The grid spacing is…

Done P2L32

- p. 4, l. 2: And which RFI filter are you using? Either write it here or refer to where you describe this in the paper.b

Added a reference here to Section 3.2 containing the description of the RFI filtering method. P4L5

- p. 5, l. 11-13: First, you write about "bias" and "RMSD", then about "bias" and "standard deviation". Are RMSD and standard deviation the same here?

They are all RMSD. We modified the text for consistency.

- p. 6, l. 3 : Reference to "Eq. 3" is misleading here. Eq. 3 has not been given yet and could mean that you refer to Eq. 3 in Zhao et al. (2015). Also at p. 6, l. 8: First give Eq. 3, then refer to it.

Modified the section to solve the issues raised. Removed the reference (P6L10) at the start of the paragraph which might indicate it's part of of the Zhao paper, and we moved the equation before any reference to it. P6L15-25

- p. 6, l. 8-18: It sounds like C is determined first and then only the other parameters (ah, bh,...) are determined? Is this / how is this done? Are ah, bh, ... determined differently than C?

For each iteration both C and the ah, bh,... group are determined. C is computed first (at each iteration) because is used in Eq. 3 to determine the other parameters. Since C/2 is considered to be the intensity at nadir, and the other parameters basically describe the two curves (Tbh, Tbv) as a function of incidence angle, C/2 will represent both Tbh and Tbv at 0 deg incidence angle.

"Are ah, bh, ... determined differently than C?" - it is explained in the paragraph previous to Eq. 3, that "C (Eq. 3) is determined for a given grid cell by first summing up the brightness temperatures of horizontal and vertical polarization for each individual observation and then taking the median of the result." and "The other five fit parameters ah , bh , av , bv and dv in the fit functions (Eq. 3) are determined by a least squares procedure." In short: at each iteration the C is obtained by summing up the TBs below 40 deg incidence angle and taking the median, while the rest of the parameters are determined by using a least square procedure. Both steps are done at each iteration. P6L15-26

- p. 6, l. 19-20: To make this clearer, I suggest to add something like: In this case, the least squares method to fit the parameters is repeated.

We added another sentence to make it more clear that after data removal the all the fit parameters (including C) are recomputed. P6L27-30

- p. 6, l. 8-24: It would be helpful to clearly state that you do not take into account the measurements that do not meet the defined criteria after a maximum of five iteration steps (if this is how these cases are handled...).

The measurements that still don't meet the criteria after five iterations are kept. The higher RMSD that these fits will have will result in higher uncertainty for the SIT computation. The fit measurements that are discarded are the ones that do not achieve convergence during the least square procedure. A new paragraph was added to make these things clear. P6L31-P7L2

- p. 6, l. 28 & p. 7, l. 4: You probably refer to Fig. 1 here instead of Fig. 2.
Yes. Issue solved. P7L8, P7L16

- p. 7, l. 20: "This is ..." -> What is "this" here? (bias and/or RMSD?)
Both of them. We changed the sentence to make it more clear. P7L30-P8L2

- p. 7, l. 20- 23: Complicated/unclear description of what you do here... E.g. "selecting all grid cells with that thickness..." -> what is "that" thickness?

For the daily average algorithm, we select all grid cells that contain SIT from each thickness bin (1 cm, 2 cm, 3 cm, ...) and then we compare them with the thicknesses from the same grid cells computed using the TB fitting procedure. This way we can compute the bias (relative to the daily average algorithm) and RMSD for each 1 cm bin, also we can look at the distribution of averaged differences along the thickness axis instead of a averaged difference that includes all thicknesses, since we expect higher differences at the higher thicknesses. This way we can also see if there is any change in sign of the averaged difference between the low and high thicknesses. We modified this part to make the procedure more clear. P7L30-P8L2

-p. 7, l. 22-23: " Only grid cells that contain at most 50 cm and non-zero in at least one of the two algorithms are used." -> Make it clearer whether the condition "in at least one of the two algorithms" applies also to the selection of grid cells with at most 50 cm SIT.
It doesn't apply to the 50 cm threshold. Both retrieval need to have a maximum of 50 cm since this is the upper limit that we consider for the algorithm. While the lower limit where we require that at least one algorithm contains sea ice higher or equal to 1 cm, so that we eliminate the areas where the algorithms detect just open water, making the bias and RMSD really small due to the large amount of open water pixels. We split the phrase in two sentences to reflect this. P7L33-P8L1

- p. 7, l. 27 & l. 28: "always generating" & "will generate" -> For clarity, maybe add "falsely"/'spuriously".
Sentences modified. P8L5-6

- p. 7, l.35: What is the "absolute bias" (as compared to just "bias"?)
The sentence has been removed since the RMSD conveys better the deviation from the mean. The "absolute bias" is computed as the normal bias where the differences between the data point and the expected values are computed as a modulus.

- p. 9, l. 6: Does the reference to "(Sect. 4.1) refer to Sect. 4.1 in Huntemann et al. (2016) or what is meant here?
All the Section references in the manuscript are just in-paper pointers, thus it refers to Section 4.1 of the current manuscript.

- p. 11, l. 1-2: "a weighted standard deviation... is used" -> weighted for what?
Distance. When we grid the SMAP TBs to a grid we use a nearest Gaussian weighting where the weight for each individual data point that goes into computing the value has a weight equal with exp(-dist**2/sigma**2) where sigma is FWHM/2(log2), with FWHM of SMAP taken as 40000 meters. We added Eq. 5 and its description to make this more clear. P13L13-18

- p. 11, l. 3: "The correlation was calculated for a period of seven days." -> Why only seven days?
Since we changed to Q,I space instead of TBh and TBv, the correlations have been recalculated (for Q and I) for a period of 3 months (1 Oct. - 31 Dec. 2015) P13L18-19

- p. 11, l. 22-24: Not very clear description.

For each pair of TBh and TBv that were used in the first step to compute SIT, we use Eq. 5 to change the TBs relative to a desired SIC. The TBs computed using the mix between sea ice (the TBs used in the first step) and open ocean TBs (the tie points) are decreased. The higher the SIC the higher the decrease, as expected. Then we compute the SIT containing the SIC influence.

At this point we group the initial SIT in bins of 1 cm (from 0 to 50 cm), and then for each bin, and each set of SIC(15, 30, 50…), we take the corresponding retrieved SIT from the second step and average them, thus to obtain one value of SIT, influenced by SIC, for each cm of SIT computed with the 100% SIC assumption.

Sentences modified. P12L1-4

- p. 11, l. 30-31: "As we have observations at two polarizations at each grid cell available, it should in principle be possible to retrieve SIT and ice concentration simultaneously." -> "In principle", this is only possible if TB varies independently with ice concentration and ice thickness, which would have to be shown before stating phrases like this.

We changed the phrasing so it can't be interpreted as a statement. P12L110-11

- Sect. 5.2: Why are these considerations not used to estimate SIT uncertainty?

The error of SIC in the training data is now considered for additional uncertainty in the retrieval see (new) Sect. 5.2.

- throughout the paper:

a) Use "an" instead of "a" before abbreviations RFI, RMSD, L-band.

Solved.

b) The text still contains some typos / language issues (e.g. mixed usage of singular and plural).

c) TB is introduced as abbreviation for brightness temperature, SIT for sea ice thickness, but sometimes the authors use these abbreviations, sometimes not (can even vary within one sentence).

After introduction of TB abbreviation we modified all the following words to be consistent.

-Is the area used for training and inter-calibration of SMOS and SMAP data defined somewhere in the paper?

The three training areas are the same as in Huntemann et al., (2014). The first paragraph of Sect. 3.1 was modified to also give the location of the three areas used for training.

For inter-calibration we use all the grid cells in the Arctic above 55 deg N. The paragraph starting at P8L31 is modified to make this clear.

-no (clear) references for statements on p. 1, l.15; p. 2, l. 4-8; p. 3, l. 5-8 & l. 13 & l. 14 & 16-19.

p. 1, l.15 Reference added. P1L16

p. 2, l. 4-8 Reference added. P2L9-10

p. 3, l. 5-8 Reference added. P3L9

[revised manuscript text omitted]
 $\cancel{\text{of}}$ as SIT as a function of $\cancel{TB_h \text{ and } TB_v}$ $Q$ and $I$ seen in Fig. $\cancel{7}$8 center and right, respectively. $\cancel{\text{Almost all of the data points will be found above the one-to-one line where the polarization difference is positive. For most of }TB_h\text{s below 200 K}}$ For $Q$ values below the 20 cm line the $\cancel{\text{rate of change of the SIT is below}}$ change rate is below 0.25 cm per K due the thickness isolines being parallel with the $Q$ axis thus for the same value of the intensity, a large change in $Q$ will result in a similar thickness value. For thicknesses between 20 and 40 cm the change increases to 0.5 cm per $\cancel{\text{K and is increasing sharply with increased }TB_h\text{ at over 230 K}}$$TB_v$. $\cancel{\text{In contrast, the derivative for }TB_v\text{ (
[revised manuscript text omitted]

---

## Author Response (AR3)

**Editor Decision: Publish subject to minor revisions (review by editor)** (19 Nov 2018) by Ted Maksym
Comments to the Author:
Based on the revised manuscript and your response to the last round of reviewer comments, I believe the manuscript is suitable for publication with a few minor corrections. I have listed some technical (mostly copy-editing corrections below, with a few small suggestions to help make the text clearer).

There are a couple areas where I believe minor changes to the text could still better address some of the reviewers' concerns. The major concerns raised by the last round of reviews were that the improvements were relatively minor and it wasn't clear why a combined product was useful. This is improved in the conclusion, but it would be helpful to the reader to know what improvements this product has in the abstract (e.g. marginally greater coverage, improved RFI, reduced TB RMSD relative to a prior merged product, etc).

Added to the abstract (P1L9-16): "The new merged SMOS/SMAP thin ice thickness product improved in several ways compared to previous thin ice thickness retrievals: (i) The combined product provides a better temporal and spatial coverage of the polar regions due to the usage of two sensors. (ii) The RFI filtering method was improved, which results in higher data availability over both ocean and sea ice areas. (iii) For the inter-calibration between SMOS and SMAP brightness temperatures the root mean square difference (RMSD) got reduced by 30% relative to a prior attempt. (iv) The algorithm presented here allows also for separate retrieval from any of the two sensors, which makes the ice thickness dataset more resistant against failure of one of the sensors. A new way to estimate the uncertainty of ice thickness retrieval was implemented, which is based on the brightness temperature sensitivities."

Second, was the lack of in situ validation. You have addressed this in section 6 as I suggested. It does seem to show some relationship, although a lot of scatter. It is obviously a limited comparison, and not terribly conclusive. Certainly there is too much scatter to say anything about whether your algorithm is any better than any prior algorithm. The manuscript could benefit from a few sentences further discussion on this point. First, how does this compare to any prior published validation of SMOS and/or SMAP against in situ data? What does this new comparison say in general about the validity of SMOS/SMAP SIT (you do comment on this a bit)? Second, I think this is not so much the point, as you have shown that the new combined algorithm compares well to a previously validated single sensor product – so you should say this explicitly in the text (I think this is what your last sentence of the conclusions is trying to say, but it is not clear as written). Lastly, you do acknowledge Schmitt and Kaleshke (2018), but say nothing about it. You should explicitly discuss any similarities or differences between your results and theirs in your discussion.

We also changed the conclusion to make it more clear that the validation of the original product should still hold for the new one (P17L13-16).
We included a comment to the end of the validation section (P16L17-27) to answer the first point, regarding the SMOS SIT validation attempt presented in Kaleschke et al., (2016).
We included comments regarding Schmitt and Kaleshke (2018) in the different relevant sections:

3.2 SMOS TBs fit characteristics
"A similar approach for fitting SMOS L1C TBs to a fixed incidence angle using the method presented in Zhao et al. (2015) was done in Schmitt and Kaleschke (2018). For filtering RFI it uses RFI flags within the SMOS data, similar to what is done in this study. As a second step, however, they remove whole snapshots if one data point within the snapshot contains a TB value over 300 K. This was also done previously in Huntemann et al. (2014). For this study, however, we introduced an iterative method to fit the brightness temperatures, which does not need a fixed cut-off value for brightness temperature removal anymore. As a result, more data will be removed before the fitting procedure in Schmitt and Kaleschke (2018) compared to the method presented here." (P7L13-19)

4.1 SMAP/SMOS inter-calibration

"In Schmitt and Kaleschke (2018) a similar comparison is done to represent the differences between the SMOS and SMAP TB datasets. Compared with the inter-calibration done here, two years of data is used instead of three month, covering also the summer period over the Arctic Ocean. Since we consider that the algorithm presented here is valid just during the winter period, a calibration that covers summer months is not necessary. The RMSD between the SMOS and adjusted SMAP TBs in Schmitt and Kaleschke (2018) are between 1 to 3 K, which is in the same range of values presented in this paper, i.e. 2.7 and 2.81 K for T B h and T B v , respectively." (P10L19-24)

"For comparison, the bias and RMSD between SMOS and SMAP SIT found in Schmitt and Kaleschke (2018) are 1 cm and 7 cm, respectively, which is slightly larger than the results presented here. However, the time period considered in Schmitt and Kaleschke (2018) is different and the SIT retrieval is based on Tian-Kunze et al. (2014), thus having a different underlying principle." (P10L31-34)

Technical and other minor corrections:

P1L9 - "sea ice thickness" lower case (P1L9)
P1L20 "multi-angular" also "organized in approximately" (P2L2)
P1L22 – insert comma after "(Font et al., 2010)" (P2L4)
P2L6 – "on board" (P2L14)
P2L10 – "radio frequency interference" lower case (P2L18)
P3L22 – "at 6 pm" (P3L27)
P3L29 – "surface based" (P4L4)
P4L4 – "The new data version exhibits a" (P5L10)
Solved.

P5L10 – As previously pointed out by a reviewer, you use bias to describe the difference between your version and the Huntemann et al. (2014) version. Of course, we don't know which is more correct, so bias is not the best term. Perhaps just say the the RMSD is warmer in the new version.
In this specific case we use the term bias, as it is used in the release notes for the 6.20 SMOS L1C data version. Both, the old and the new data versions were compared (SMOS Calibration team and Expert Support Laboratory Level 1, 2015) with an ocean forward model resulting in a bias of the new data of approximately 1.5 K relative to the model. This bias was specified also in Section 2 P3L19-26, while now Fig. 2 show an example in the TB change between the data versions. In the previous iteration of the manuscript we changed the term "bias" to "difference" in most instances where we compared the new TB and SIT results with the old ones but for this instance we consider that the term bias is appropriate.

P5L13 –This sentence is not very clear. I think you mean "The spillover produces an erroneous increase in TB over ocean areas adjacent to coastlines or ice edges, or decreases in TB over sea ice near the ice edge."
Phrase modified as suggested. (P5L17)

P5L16 – please state what you mean by spatial ripples.
Added: "originating from the Fourier reconstruction of the snapshot" and reference to Corbella et al., (2005). (P5L21)

P5L20 – "at most a 3 cm difference"
Solved. (P5L26)

P5L27 – be careful here calling this the error. You of course don't know what the real error is, only that there is a difference of <1 cm thickness between the two algorithms.
Changed "error" to "difference". (P5L32)

P7L4 – change "As opposite" to "conversely"
Solved. (P6L10)

P7L18 "Due to the fit computational requirement to have observations below 40°"
Changed to "Due to the requirement of the fitting procedure to have observations below 40°" (P7L32)

P7L20 "The decrease is around 1° in latitude" – this is not so clear. Where geographically is this? Do you mean around the pole? If so please indicate this.
Rephrased to: "The decrease of the covered area surrounding the North Pole, relative to the old algorithm, is around 1° in latitude, corresponding to approximately 1000 grid cells." (P7L33-P8L1)

P7L23 "around Iceland, Eastern Greenland and Vladivostok" – I do not understand why this is a positive point. None of these areas have sea ice at this time of year (unless you mean northeast Greenland, but I do not think you do, and Iceland never has sea ice). I do not see how this is an advantage. Any retrievals here are clearly in errors. I recommend either indicating that there are increased errors in some coastal areas (although these could be flagged by other data sources, such as AMSR2) or removing this discussion
We did mean north eastern Greenland since there is an area that is not covered by the old algorithm but by the new algorithm. We removed the other two mentioned areas (P8L3) which are cover by open water and kept just North Eastern Greenland. This set of figures was presented just to give an example of improvements of coverage overall. The old algorithms 2010 winter SIT data is affected by RFI, especially in in the area mentioned. We used this specific date because it contains both a sea ice RFI affected area and also a large area north of Siberia with thin sea ice, so that we can give a one day example of difference between the old and new method.

P7L27 – "Figure 3, right" not Figure 4. (P8L6)
P7L31 – remove extra "the" (P8L12)
P7L33 – change "with that thickness" to "with thicknesses falling within that bin" (P8L14)
P8L3 – change "Until 40 cm of thickness" to "for ice less than 40 cm thick" (P8L18)
P8L5 change "values is below" to "ice thickness differences are below" (P8L20)
Solved.

P9L14 – you use bias here again to describe the difference between the SMOS data versions. As pointed out previously by a reviewer, this term is imprecise when used to refer to a difference between the algorithms when neither had been clearly validated against ground truth or other independent data (I understand the prior version has been compared to data in previous work, but it's not immediately clear that version 6.20 compares any worse as this is not shown). I recommend here, and throughout the manuscript, that where you have used "bias" to describe a difference between two datasets where there is actual true value is not also known or compared to, that you use something like "difference" instead. If you do know that the prior version has been validated with zero bias against an independent data source, then it would be fine to use bias in your context, but only if you make it clear that the prior data set had zero bias and relative to what data (if the prior version had a bias of its own, then you would need to compare these to be able to define what your new bias is relative to).
As already explained for the P5L10 comment, the term bias regarding the 6.20 SMOS data version is used as presented in the release notes of the new data version relative to tests done by the SMOS Calibration team relative to a forward model. The higher TBs of SMOS over both ocean and sea ice is seen in the comparison of the two data

version, and between the new SMOS data and the SMAP data as presented in this paragraph. We consider that the term bias, in this specific case, is correct.

P9L20-34 This section is not as clear as it could be. You jump between steps used in your algorithm and procedures used by Huntemann, so it is not clear to the reader if a particular sentence refers to your work or Huntemman's. e.g. the first sentence on lines 24-26 appear to refer to your study, while the second refers to Huntemman, but by jumping between it becomes unclear. It would be better if you described what Huntemann did in its entirety first, and then any limitations (so as to let the reader know why you undertook your work), and then described your procedure after.
This paragraph has been changed and divided in two to eliminate the confusion and separate better the Huntemann et al., (2016) paper and the current one. (P9L33-P10L18)

P9L33 – "both polarization, at least 1.3K higher than the one presented in this paper". Also it is still a little unclear what you are saying here. Since this seems to be one of the main selling points of the paper (i.e. there is some advantage with respect to the Huntemann algorithm), please make sure your meaning is clear here. For example, it is not clear if you mean that the improvement in the RMSD is due to the calibration to compensate for ET contributions and warm "bias", or if these are separate points. Please clarify this. It would also be useful to indicate here what the consequences of this reduced RMSD are for SIT retrieval – roughly how much improvement in SIT might you expect for a reduction of 1.3K in RMSD. This is again mentioned in the discussion, so clearly one of your main points, but you do not relate this to the SIT retrieval. I think it would be helpful to understand the significance of your paper to discuss this result in terms of its expected impact on SIT retrieval error, or quality (at least in the discussion).
Since the conversion of SMAP TBs into SMOS equivalent TBs is done through a linear regression, this doesn't differentiate between the different sources of error or uncertainty that influence either SMOS or SMAP data. Also, the retrieval is empirical and based on SMOS data and is retrained in this paper, this means that even if the TBs might be warmer than a forward model output in SMOS Calibration team and Expert Support Laboratory Level 1: SMOS L1OPv620 release note, we care that the SMAP TBs are just closer to SMOS ones.

Changed sentence to "As a consequence, the transition from SMAP to SMOS TBs requires now just one linear regression compared to Huntemann et al. (2016). In this linear regression between the revised SMOS and the SMAP TBs (Sect. 4.1 and 4.2), the RMSD reduced by more than 1.3 K, approximately 30%, compared to Huntemann et al. (2016), indicating a better match of SMOS and SMAP based brightness temperature. This, in turn, ensures smaller differences between the retrieved SIT of both instruments and allows the combination of the two retrievals into a joint SIT product." (P10L13-18)

P10L4-5 – this is an awkward sentence. Perhaps rephrase: "For the period from October 1 to December 31, 2015, the difference SITs between SMOS 40° incidence angle fitted TB retrieval and SMAP retrieval are small. Using only grid cells containing SIT < 50 cm and at least one of the two retrievals having SIT > 0 cm, the average difference for the SMOS SIT relative to the SMAP SIT is -0.2 cm and a 2.39 cm RMSD."
Phrase changed to the one proposed. (P10L27-30)

P10L14 You have previously defined SIT as sea ice thickness, but you jump between the acronym and full wording throughout. Be better to define once and use SIT throughout.
Solved.

P10L14-15 "which has greater coverage" (P11L7)
P10L19 – "this result means" (P11L12)
P10L20 – "is on average" (P11L12)

Solved.

P10 L21 – be better here, and throughout to simplify dates, i.e. October 24, 2015, instaed of 24th of October, 2015.
We changed all dates to British english standard, e.g. 24 October 2015.

P10L21 – "The greatest differences" (P11L14)
P10L22 "to over 50 cm" (P11L15)
P10L24 – Figure 6 (right panel)" (P11L16)
P10L25 "the average" (P11L17)
P10 L29 – "the data are limited to the region" (P11L22)
Solved.

P10 L32 – what fit parameters? Do you mean for equation (3)? Since the main text is a bit dense for the casual reader, it would be better if each of the steps in this section where more specific as to what equation, or part of the paper the step is referring to. More examples of this below:
Yes, we use the procedure presented in Sect. 3.2, including equation (3). We added to most bullet points the relevant in paper references.

P11L7 – "the gridded SMAP TBs are converted to SMOS equivalent TBs" – how? By linear regression to SMOS TBs.
Changed to: "the gridded SMAP TBs are converted to SMOS equivalent TBs by linear regression (Sect. 4.1)" (P12L1)

P11L9 – "are computed by error propagation" (P12L3)
P11L12 – "Polarization difference (Q) and Intensity difference (I) are" (P12L5)
P11L11 "are calculated from the" (P12L5)
Solved.

P11L15 "the sea ice concentration (SIC) and cumulative freezing degree day (CFDD) error are included in the SIT uncertainty resulting from the previous step" I am not sure this (and other error calculations) should be listed as steps in the algorithm. My impression was the reviewer requested this list of steps so that the reader could see how the algorithm works, not every analysis step you performed in the paper. The error analysis I believe is to demonstrate the effectiveness of the algorithm and its sensitivity to errors, but this is not a step in the algorithm. P11L16 this step should be deleted. I don't see this as a necessary step to the algorithm.
Bullet points removed.

P11L17 what do you mean by "them"?
Changed to: "both SMOS and SMAP". Here we mean that the process used to obtain the combine SIT can be also used for computing SIT and SIT uncertainty for each sensor separately, without the steps necessary for obtaining the mixed TBs, thus the end result for a day will be a three SIT datasets and their associated uncertainties. (P12L9-10)

P12L1 – be consistent with date format
P12L24 – "SIT below 20 cm thickness" (P13L17)
P13L30 "we take one day" (P14L23)
P14L1 – do you mean figure 9 here? (P14L26)
P14L13 "due to incorrect representation of the total.." (P15L3)
P14 L13 "due to the greater exchange of heat" (P15L4)
P14L14 "due to greater exchange of heat" (P15L6)

P14L20 again make sure you are consistent with date formats throughout the paper.
P14L32 "From the initial formation of sea ice…" (P15L21-22)
Solved.

P15L4 – include a reference for the ASPeCT protocol.
Added reference to Worby and Ackley (2000). (P15L28)

P15L4 – "During the day, this allowed for an estimate of ice thickness in an approximate radius of 1 km". (P15L28)
P15L7 "ship-based" (P15L31)
P15L11 – "results in a slope" (P16L3)
Solved.

P15L14 Fig ?? – which figure are you referring to here?
It is Figure 7. We corrected the problem. (P16L6)

P16L10 – what do you mean be "refer to their comparisons" do you mean for the reader to refer to their validation of their product?

[revised manuscript text omitted]